# Ceruloplasmin replacement therapy ameliorates neurological symptoms in a preclinical model of aceruloplasminemia

Alan Zanardi[1], Antonio Conti[1], Marco Cremonesi[1], Patrizia D'Adamo[2], Enrica Gilberti[3], Pietro Apostoli[3], Carlo Vittorio Cannistraci[4,5], Alberto Piperno[6,7], Samuel David[8] & Massimo Alessio[1,*] (iD)

## Abstract

Aceruloplasminemia is a monogenic disease caused by mutations in the ceruloplasmin gene that result in loss of protein ferroxidase activity. Ceruloplasmin plays a role in iron homeostasis, and its activity impairment leads to iron accumulation in liver, pancreas, and brain. Iron deposition promotes diabetes, retinal degeneration, and progressive neurodegeneration. Current therapies mainly based on iron chelation, partially control systemic iron deposition but are ineffective on neurodegeneration. We investigated the potential of ceruloplasmin replacement therapy in reducing the neurological pathology in the ceruloplasmin-knockout (CpKO) mouse model of aceruloplasminemia. CpKO mice were intraperitoneal administered for 2 months with human ceruloplasmin that was able to enter the brain inducing replacement of the protein levels and rescue of ferroxidase activity. Ceruloplasmin-treated mice showed amelioration of motor incoordination that was associated with diminished loss of Purkinje neurons and reduced brain iron deposition, in particular in the choroid plexus. Computational analysis showed that ceruloplasmin-treated CpKO mice share a similar pattern with wild-type animals, highlighting the efficacy of the therapy. These data suggest that enzyme replacement therapy may be a promising strategy for the treatment of aceruloplasminemia.

**Keywords** aceruloplasminemia; ceruloplasmin; enzyme replacement therapy; neurodegeneration with brain iron accumulation
**Subject Categories** Genetics, Gene Therapy & Genetic Disease; Neuroscience

## Introduction

Aceruloplasminemia (Acp) is a rare autosomal recessive disease (estimated prevalence 1 out of $2 \times 10^6$ in non-consanguineous marriages in Japan) caused by mutations in the gene encoding for ceruloplasmin (Cp) that result in the absence or in an inactive form of the protein (reviewed in Kono, 2012; Miyajima, 2015a). Cp is a multicopper oxidase glycoprotein that coordinates six copper atoms necessary for its three-dimensional structure and enzymatic activity, but is apparently not involved in copper metabolism (Gitlin et al, 1992; Musci et al, 1999; Zaitsev et al, 1999; Hellman & Gitlin, 2002; Hellman et al, 2002; Sedlak et al, 2008; Squitti et al, 2008). Cp is a ferroxidase (transforming $Fe^{2+}$ to $Fe^{3+}$) that plays a critical role in iron homeostasis by the oxidation and mobilization of iron from stores and the subsequent incorporation of ferric iron into transferrin that became available for cellular uptake via transferrin receptor (Patel & David, 1997; Hellman & Gitlin, 2002; De Domenico et al, 2007; Olivieri et al, 2011; White et al, 2012; Musci et al, 2014). The ferroxidase activity of Cp also maintains $Fe^{2+}$ efflux from cells through the ferroportin exporter (Jeong and David, 2003) that, in the absence of Cp is internalized and degraded, resulting in intracellular iron accumulation (De Domenico et al, 2007; Kono et al, 2010; Olivieri et al, 2011; Persichini et al, 2012; Musci et al, 2014). In addition, Cp has antioxidant properties preventing the production of deleterious reactive oxygen species via the Fenton reaction (Hellman & Gitlin, 2002). There are two isoforms of Cp generated by alternative splicing, a secretory form and a glyco-sylphosphatidylinositol (GPI) membrane-anchored isoform (Patel et al, 2000; De Domenico et al, 2007; Marques et al, 2011, 2012). Cp is predominantly synthesized by hepatocytes and secreted into the blood, but extrahepatic expression has been demonstrated in several tissues, including central nervous system (CNS; Hellman & Gitlin, 2002). Within CNS, Cp is expressed as both the soluble isoform

1   Proteome Biochemistry, Division of Genetics and Cell Biology, IRCCS-San Raffaele Scientific Institute, Milan, Italy
2   Molecular Genetics of Intellectual Disabilities, Division of Neuroscience, IRCCS-San Raffaele Scientific Institute, Milan, Italy
3   Unit of Occupational Health and Industrial Hygiene, Department of Medical and Surgical Specialties, Radiological Sciences and Public Health, University of Brescia, Brescia, Italy
4   Biomedical Cybernetics Group, Biotechnology Center (BIOTEC), Center for Molecular and Cellular Bioengineering (CMCB), Department of Physics, Technische Universität Dresden, Dresden, Germany
5   Brain Bio-Inspired Computation (BBC) Lab, IRCCS Centro Neurolesi "Bonino Pulejo", Messina, Italy
6   School of Medicine and Surgery, University of Milano Bicocca, Monza, Italy
7   Centre for Diagnosis and Treatment of Hemochromatosis, ASST-S.Gerardo Hospital, Monza, Italy
8   Center for Research in Neuroscience, The Research Institute of The McGill University Health Center, Montreal, QC, Canada
    *Corresponding author. Tel: +39 2 26434725; E-mail: alessio.massimo@hsr.it

secreted into the cerebrospinal fluid (CSF) by epithelial cells of the choroid plexus, and a membrane-bound isoform expressed by different cells, including astrocytes and retinal pigment epithelium (Klomp *et al*, 1996; Patel & David, 1997; Patel *et al*, 2000; Mittal *et al*, 2003). Iron levels in the brain are tightly regulated via Cp and other related proteins to minimize the effects of systemic iron deficiency on brain function and to protect CNS from iron-mediated free-radical injury (Patel *et al*, 2002).

Aceruloplasminemia is caused by mutations in CP gene leading to absent or dysfunctional protein. So far, ~50 pathogenic variants of Cp have been identified worldwide in more than 60 families from different racial groups, even though the larger number of families has been identified in Japan (Kono, 2012; Miyajima, 2015a). In humans with Acp, the absence of Cp ferroxidase activity leads, on one hand, to a decreased iron delivery to transferrin and iron-restricted erythropoiesis, and, on the other hand, to iron accumulation in the parenchyma of several tissues including liver, pancreas, heart, thyroid, retina, and brain (Kono, 2012; Miyajima, 2015a). Typical Acp manifestations are mild microcytic anemia and complications due to iron-mediated cytotoxicity. The latter, fostered by iron deposition, results in retinal degeneration, diabetes, hypothyroidism, and cardiac failure that generally precede by about 10 years the onset of neurological and psychiatric symptoms (Fasano *et al*, 2007; Kono, 2012; Miyajima, 2015a; Vroegindeweij *et al*, 2015b). The clinical manifestations at diagnosis are heterogeneous and range from a full-blown phenotype characterized by anemia, tissue iron overload, and iron-related complications, to only iron-restricted anemia and tissue iron overload in the absence of non-hematological manifestations. For example, it has been reported that in 71 Acp patients, only 68% displayed neurological signs, while 80, 76, and 70% showed anemia, retinal degeneration, and diabetes, respectively (Miyajima, 1993–2017). Although different Cp mutations and genetic backgrounds might be implicated, data are not sufficient to draw conclusion explaining phenotype heterogeneity.

Aceruloplasminemia is a fatal disease with a long-lasting neurological disability that reflects brain iron deposition, and for this reason, it is placed in the group of disorders known as neurodegeneration with brain iron accumulation (NBIA; Kono, 2012; Vroegindeweij *et al*, 2015a). The iron deposition and iron-mediated free-radical stress found in Acp are thought to be responsible for astrocyte and neuronal cell death (Miyajima *et al*, 1996; Yoshida *et al*, 2000; Kono, 2012; Miyajima, 2015a). Neurological symptoms are the predominant clinical feature and usually manifest in the fifth–sixth decade of life progressively leading to cerebellar ataxia, dyskinesia, parkinsonism, depression, psychiatric changes, and dementia (Kono, 2012; Miyajima, 2015a; Vroegindeweij *et al*, 2017). Current therapies mainly based on iron chelators, partially control systemic iron deposition but are poor or ineffective in controlling neurological symptoms (Dusek *et al*, 2016). Moreover, side effects (e.g., increased anemia) limited long-term chelation therapy that would be required to mobilize iron from the brain (reviewed in Kono, 2012). In few cases, chelation therapy was empirically combined with fresh-frozen plasma (FFP) administration in order to restore both the blood iron and Cp levels, but only two case reports suggested temporary beneficial effects (Yonekawa *et al*, 1999; Poli *et al*, 2017). Thus, there is a need to scientifically prove that parenteral Cp administration can be a treatment option for Acp patients.

Cp-knockout (CpKO) mouse models show several features of Acp in humans (hepatic iron overload, mild anemia, low serum iron, and increased ferritin), normal copper metabolism, and pathological effects secondary to iron dysmetabolism and iron-related toxicity (Harris *et al*, 1998; Patel *et al*, 2002; Jeong & David, 2006). In the C57Bl/6J genetic background, CpKO mice also develop neurodegenerative phenotype with aging, due to iron deposition in the cerebellum and brainstem, increased lipid peroxidation, and iron-mediated damage (Patel *et al*, 2002; Jeong & David, 2006). These mice showed deficit in motor coordination that were associated with a loss of brainstem dopaminergic and cerebellar neurons (Patel *et al*, 2002; Jeong & David, 2006). In addition, young CpKO mice were more susceptible to oxidative stress and neurotoxicity, supporting the antioxidant role of Cp (Patel *et al*, 2002; Kaneko *et al*, 2008; Hineno *et al*, 2011). In the CpKO mice that show early neurological defects at 6–8 months of age (Jeong & David, 2006; Ayton *et al*, 2013), it has been reported that parenterally administered Cp is able to enter the brain (Ayton *et al*, 2013).

Here we show the therapeutic potential of Cp replacement in reducing the neurological pathology in the CpKO mice model of Acp. We found that intraperitoneally administered human Cp is able to cross the brain-barrier systems inducing, after 2 months of treatment, a partial replacement of the Cp levels, rescue of brain ferroxidase activity, and amelioration of motor incoordination. This was associated with both diminished neuronal loss and reduced brain iron deposition. Multivariate dimensional reduction analysis showed that CpKO mice treated with Cp were similar to the WT animals, supporting the efficacy of Cp replacement therapy for Acp.

## Results

### Intraperitoneally administered Cp enters the brain of CpKO mice and exerts its ferroxidase activity

In order to assess whether repeated intraperitoneal (IP) administration of Cp can result in Cp accumulation in brain, we performed pharmacokinetics experiments on 3-month-old CpKO mice that were treated twice IP with Cp, and the second injection was made 5 days after the first. Western blot analysis on brain tissue homogenates showed that the IP-administered Cp is able to reach the brain across the brain-barrier systems (Fig 1A and B). After Cp administration into CpKO mice, Cp was detected as band of about 150 kDa, appearing in addition to background signals (Fig 1A). This band was reduced over time but was maintained in brain homogenates apparently being more stable after the second injection (Fig 1A and B). In the same brain homogenates, an increase in ferroxidase activity was detected concomitantly to the Cp injection and was stable along time (Fig 1C). Since mice were perfused before being sacrificed, we can exclude any possible contamination by Cp still present in the blood that could have affected the Cp levels and the ferroxidase activity detected in brain homogenates. The results showed that repeated intraperitoneal administration allows functional Cp to enter the brain with stable levels.

### CpKO mice at 8 months of age show impaired motor coordination

In order to assess the presence of neurological symptoms in 8-month-old CpKO mice before replacement treatment was initiated, we performed behavioral tests in CpKO and WT mice ($n = 12$,

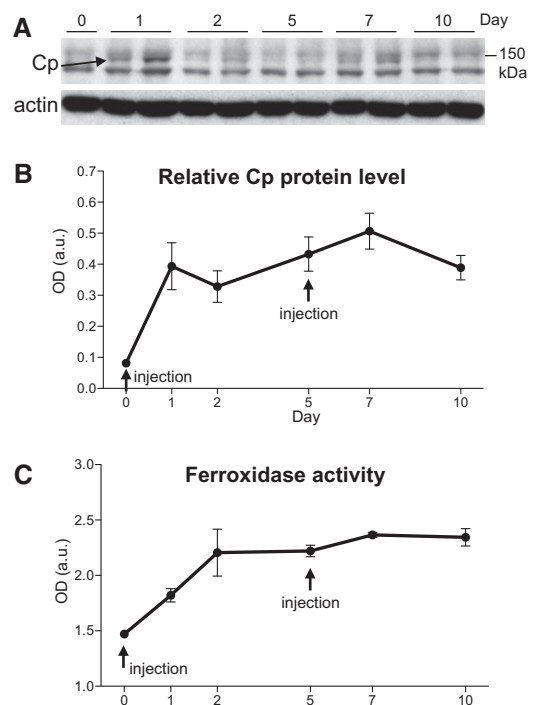

**Figure 1.  Pharmacokinetics analysis of Cp administration.**

A  Western blot showing Cp level (arrow) at the indicated time points in representative CpKO mice injected twice with purified Cp at days 0 and 5. Western blot for actin is used for signal normalization.

B  Optical density (OD) analysis of Western blot Cp signal in brain showing the kinetics of protein appearance. Cp signal is normalized for actin expression.

C  Kinetics of ferroxidase activity measured in brain along the time points of Cp administration.

Data information: In (B, C), data are presented as mean ± SEM; $n$ = 3 mice for each time point with two technical replicates.

5 males and 7 females each group). CpKO mice showed significant motor coordination impairment compared to WT animals, as assessed by the grid test and the Rotarod test, the latter performed in acceleration speed (Fig 2A and B). The impairment indicated that a neurological deficit was present in CpKO mice prior to the treatment. Analyzing accelerated Rotarod test performance in the five consecutive trials, we observed a clear different behavior between WT and CpKO mice (Fig 2C). Note that the lack of significance between the two groups at the first time point test is likely due to the novelty of the task. In subsequent trials the WT mice remain on the Rotarod for 5 min, the maximum time allowed (Fig 2C).

## CpKO mice treated with Cp show improvement in motor coordination

The behavior assays were performed at 10 months of age after 2 months of treatment with Cp (CpKO + Cp and WT + Cp; $n$ = 6, 2M, 4F each group) or with control saline buffer (CpKO and WT; $n$ = 5; 2M, 3F each group). In the groups of mice treated with saline, two mice were not included in the analysis because one CpKO mouse was killed by its cage's companions and one WT mice showed an aberrant mass, likely tumoral. CpKO mice treated with Cp, showed

an improvement of motor coordination compared to untreated CpKO animals, both on grid and Rotarod tests (Fig 2D–F). On the grid test, the performance of the Cp-treated CpKO mice was similar to the WT but not yet significantly different from the CpKO mice ($P$ = 0.063; Fig 2D). On the contrary, on the accelerated speed Rotarod tests, which evaluate a higher degree of motor coordination, the Cp-treated CpKO mice showed a statistically significant improvement compared to the untreated CpKO mice (Fig 2E). The representation of the performances in the single consecutive trials of the accelerated Rotarod analysis shows the consistency of the effect that therapeutic Cp administration had on CpKO mice (Fig 2F). Taken together, these results indicate that the therapeutic replacement of Cp was able to ameliorate the neurological symptoms of motor impairment.

## Intraperitoneally administered Cp accumulates in the brain of CpKO mice but not in WT mice

Two months of treatment with Cp induced a partial replacement of Cp levels in the brain of CpKO mice, as assessed by Western blot analysis, while at the end of the treatment, Cp was not detectable in the serum of CpKO mice (Fig 3A–C). This indicates that IP-administered Cp has a tendency to accumulate in brain tissue, while it is quickly cleared from the blood. In brain extracts, the IP-administered human Cp was clearly distinguishable from the endogenous Cp visible in the WT mice because of its higher relative molecular mass (Fig 3A, arrow vs. head arrow). A similar difference in molecular weight has already been reported between human and rat Cp (Hahn *et al*, 2004). A difference in the electrophoretic migration between endogenous mouse Cp and administered human Cp was also evident in serum (Fig 3A, arrow vs. head arrow). Apparently, none of the administered Cp was able to enter the brain of the WT animals, at least in detectable amounts as in CpKO mice, suggesting a different ability of the administered Cp to enter the CNS in CpKO mice. By comparing Western blot signals of Cp in CpKO and WT mice, and that of purified Cp loaded as control, we found that Cp accumulates in the brain of CpKO mice in variable amounts ranging from 50 pg to 2 ng in 30 μg total lysate loaded on the gel. These amounts represented 1/20 to 1/1 of the homogenous endogenous Cp signal detected in the WT mice. Probably due to this large variability, the difference in the Cp levels showed only a trend but nevertheless clearly indicated that Cp accumulated in the brain of treated CpKO mice (Fig 3B).

## IP-administered Cp does not enter in the brain by transcytosis in barrier systems

Based on previous evidence showing that Cp can cross hepatic sinusoid endothelium via coated vesicles by receptor-mediated transcytosis (Tavassoli, 1985; Irie & Tavassoli, 1986; Tavassoli *et al*, 1986), we investigated whether this was the mechanism by which Cp entered the brain of CpKO mice. Since previous studies showed that Cp is desialylated by hepatic endothelial cells during transcytosis (Tavassoli, 1985; Irie & Tavassoli, 1986; Tavassoli *et al*, 1986), we checked the sialylation status of Cp in brain homogenates of CpKO-treated mice. The administered purified human Cp is a sialylated glycoprotein as inferred by reactivity with the sialic acid-specific SNA and MAA lectins, and by the loss of reactivity after sialic acid removal by neuraminidase treatment (Appendix Fig S1). Removal of sialic acids also resulted in slight reduction in the molecular weight of

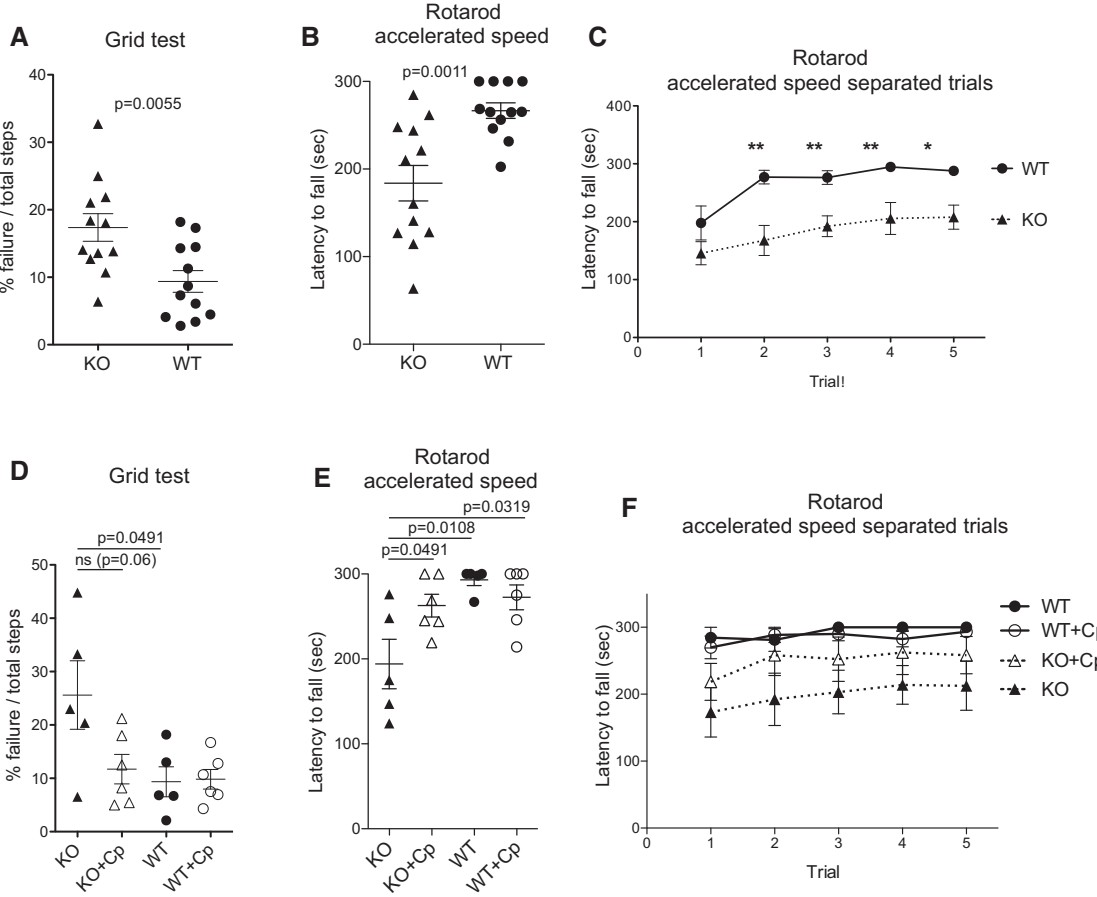

**Figure 2. Motor coordination behavioral tests before and after Cp treatment.**

A–C    Behavioral tests in mice at 8 months of age, before the Cp treatment. Grid test (A), Rotarod test at accelerated speed (B), and at accelerated speed analyzed as separate trials (C). In (A) and (B) each dot corresponds to one animal, *n* = 12 mice each groups.

D–F    Behavioral tests in mice at 10 months of age, after 2-month Cp treatment. Grid test (D), Rotarod test at accelerated speed (E), and at accelerated speed analyzed as separate trials (F). In (D) and (E) each dot corresponds to one animal (CpKO, *n* = 5; CpKO + Cp, *n* = 6; WT, *n* = 5; WT + Cp, *n* = 6).

Data information: In (A, D), data are presented as mean ± SEM of the animal groups average obtained from three trials for each mouse (Student's *t*-test). In (B, E), data are presented as mean ± SEM of the animal groups average obtained from five trials for each mice (Student's *t*-test). In (C, F), data are presented as mean ± SEM of the single trial for mice of each group. *$P < 0.05$, **$P < 0.01$ (Mann–Whitney *U*-test).

purified Cp (Appendix Fig S1), while the Cp found at the end of the treatment in the brain homogenates of CpKO mice displayed a molecular weight greater than the endogenous Cp expressed in WT animals, and similar to the original purified human Cp (Fig 3A). The same brain homogenates exhibited a reactivity of the SNA-MAA lectins corresponding to the band related to the administered Cp, not present in untreated CpKO mice (Fig 3D images on the left). Furthermore, treatment with neuraminidase resulted in a reduction in Cp molecular weight and disappearance of SNA-MAA lectins reactivity in Cp-treated CpKO mice (Fig 3D). Together, these evidences suggested that the administered Cp that reached the brain is still sialylated and entered the CNS by a mechanism different than endothelial transcytosis.

## Cp administration induces recovery of the Cp-associated ferroxidase activity in the brain of CpKO mice

Ceruloplasmin ferroxidase activity was evaluated in brain and serum of treated CpKO mice by apo-transferrin and

bathophenanthroline assays, respectively. Despite the lower protein expression compared to the endogenous Cp, Cp administration induced a complete recovery of the Cp-associated ferroxidase activity in the brain of CpKO mice. On the contrary, an unexpected non-significant trend in reduction in ferroxidase activity was observed in the brain of Cp-treated WT mice (Fig 4A). In serum, in accordance with the almost undetectable levels of Cp protein, the rescue of the ferroxidase activity in Cp-treated mice was very low (about 9% of the activity detected in the WT mice), even though the activity was significantly improved compared to the CpKO-untreated mice (Fig 4B). The addition of sodium azide in the assay, that selectively hampers Cp activity (Gray *et al*, 2009; Ayton *et al*, 2013), was able to inhibit the ferroxidase activity in Cp-treated CpKO mice and in WT mice both in brain and serum (Fig 4C), indicating that the rescue of ferroxidase activity measured in treated CpKO was due to the administered Cp.

The observation that the ferroxidase activity of the administered Cp was preserved in brain suggested that, in contrast to older CpKO

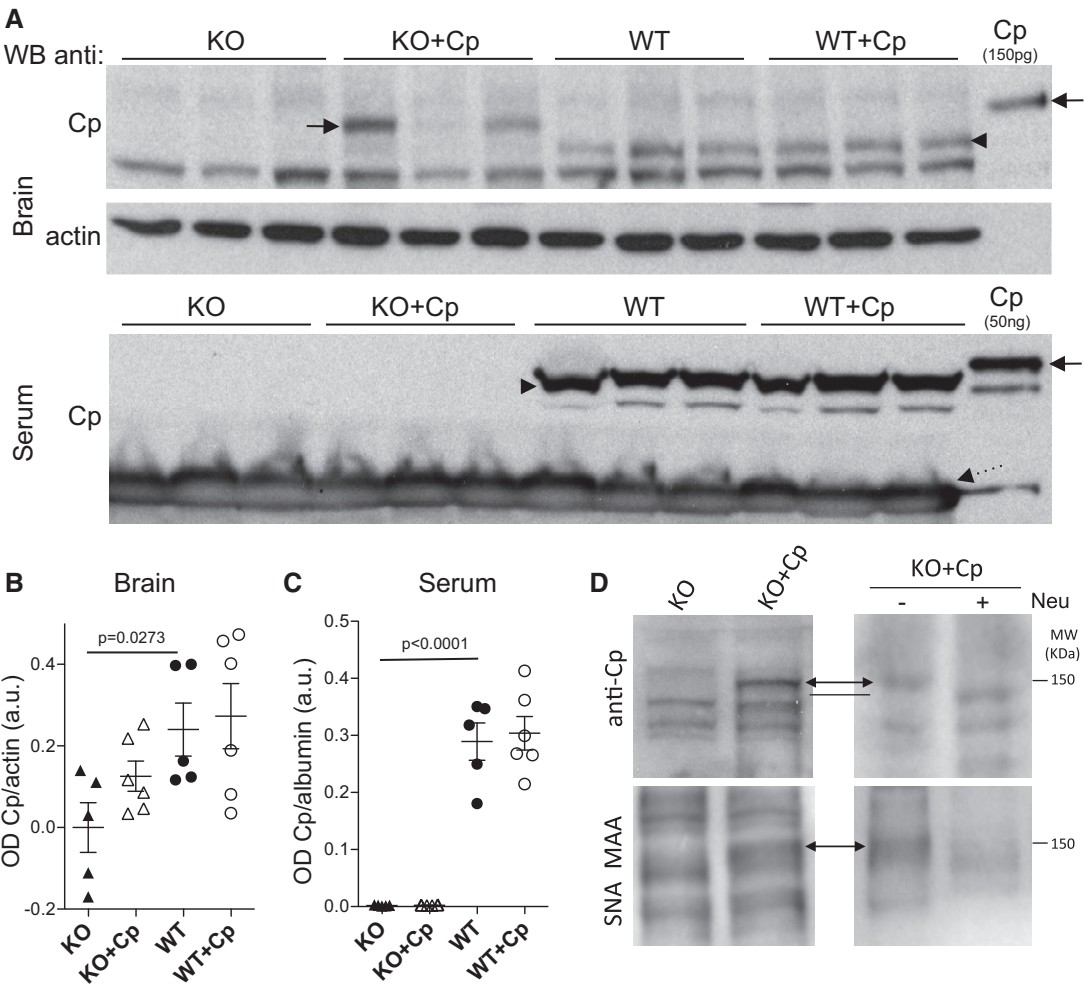

**Figure 3.  Cp-level analysis in brain and serum at the end of Cp treatment.**

A   Western blot showing Cp level in brain and serum at the end of treatment in representative animals. Arrows indicate the exogenous administered Cp, and arrowhead indicates endogenous Cp. Dashed arrow indicates albumin background signal in serum samples. Purified Cp was run as control for semi-quantitative evaluation (Cp, lanes on the right). Western blot for actin is used for signal normalization.

B   Optical density (OD) analysis of Cp level normalized for actin in brain.

C   Optical density (OD) analysis of Cp level normalized for albumin in serum.

D   Analysis of the sialylation status of administered Cp found in the brain of CpKO mice treated with Cp. Top panels: anti-Cp antibody reactivity. Bottom panels: SNA-MAA lectins reactivity. Right panels: comparison with reactivity after neuraminidase (Neu) treatment. Double arrows indicate sialylated Cp; dash indicates anti-Cp reactivity after sialic acid removal. Image contrast has been homogeneously modified to better identify the bands.

Data information: In (B, C), data are presented as mean ± SEM of the animal groups average, and statistical significance was evaluated by Student's *t*-test (B) and by Mann–Whitney *U*-test (C). Each dot corresponds to one animal (CpKO, *n* = 5; CpKO + Cp, *n* = 6; WT, *n* = 5; WT + Cp, *n* = 6). Technical replicate *n* = 3 experiments.

mice (Patel *et al*, 2002), the pathological environment was not yet pro-oxidant in mice at 10 months of age. This hypothesis was confirmed by the evaluation of total protein oxidation performed by OxyBlot assay that showed a similar total level of protein carbonylation in CpKO and WT mice regardless the treatment with purified Cp (Appendix Fig S2).

**Cp treatment elicits an anti-human Cp serological immune response that was unable to neutralize the Cp ferroxidase activity**

We evaluated the possible induction of a serological immune response against exogenous human Cp by ELISA. For this, we measured serum immunoglobulin (Ig) reactivity against purified Cp. Despite the high degree of homology between human and mouse Cp (> 80%), after 2 months of treatment, both CpKO and WT mice injected with human Cp developed an anti-Cp IgG immunological response (Fig 5A). Therefore, we purified from mice sera the IgG fraction that showed to retain the Cp-binding specificity (Fig 5B). We used purified Igs to assess whether this affected the Cp ferroxidase activity. Incubation with purified IgG from both CpKO and WT mice showed no neutralizing effect on human Cp ferroxidase activity (Fig 5C), while functional suppression/restriction was observable after Cp heat inactivation or incubation with a mixture of commercial anti-Cp antibodies (Fig 5C).

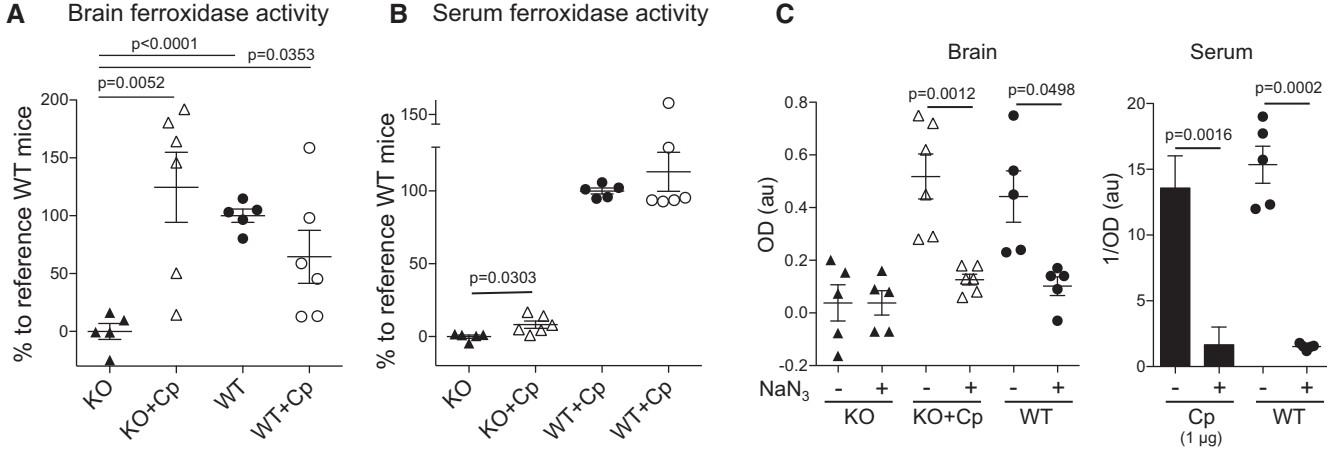

**Figure 4. Ferroxidase activity analysis in brain and serum.**

A  Ferroxidase activity in brain, measured with apo-transferrin assay, reported as percentage of the reference activity of the WT mice.

B  Ferroxidase activity in serum, measured with bathophenanthroline assay, reported as percentage of the reference activity of the WT mice.

C  Inhibition of both administered and endogenous Cp ferroxidase activity with sodium azide (NaN$_3$) in brain and serum measured by apo-transferrin and bathophenanthroline assay, respectively. Purified Cp (1 μg) was used as positive control. Data are reported as optical density (OD) or 1/OD according to the assay features.

Data information: Data are presented as mean ± SEM of the animal groups average, and statistical significance was evaluated by Student's *t*-test (A, C) and by Mann–Whitney *U*-test (B). Each dot corresponds to one animal (CpKO, *n* = 5; CpKO + Cp, *n* = 6; WT, *n* = 5; WT + Cp, *n* = 6). Technical replicate *n* = 3 experiments.

## Cp administration reduces both total brain iron content and iron deposition in choroid plexus epithelium in CpKO mice

Quantitative analysis of total iron content in brain homogenates of 10-month-old CpKO mice, performed by inductively coupled plasma mass spectrometry (ICP-MS), showed significant iron accumulation compared to WT mice (Fig 6A). Of note, Cp treatment is able to significantly reduce the amount of iron in the brain of CpKO mice to those of WT mice (Fig 6A). Cp replacement did not significantly affect the brain concentration of other metal ions like copper and zinc (Fig 6B and C). Similar to the brain, we found a significantly higher iron concentration in the liver of CpKO mice compared to WT (1,535 ± 341.3 vs. 517.1 ± 164.4 μg/g dry tissue), which appeared to be reduced after Cp administration (1,208 ± 241.8 μg/g dry tissue), however, did not reach statistical significance (Appendix Fig S3).

To investigate whether specific brain regions were involved by iron accumulation, total iron detection was performed on sagittal sections of mice brain. Similar very scanty iron reactivity was observed in brain sections, including the cerebellum and brainstem areas, of the different groups of mice (data not shown). Interestingly, differences in the iron staining were found in the choroid plexus, one of the brain-barrier systems (blood–CSF barrier; Fig 6E). Semi-quantitative analysis showed in the CpKO mice a significant intracellular iron staining of choroid plexus compared to both untreated and Cp-treated WT mice (Fig 6D and E). Moreover, the CpKO mice treated with Cp showed a significant reduction in iron staining compared to untreated CpKO mice, with iron levels between WT and CpKO mice. These results suggest that the IP-administered Cp is effective in controlling iron accumulation in the choroid plexus epithelial cells (Fig 6D and E). Interestingly, no similar iron deposition was observed in brain microvascular endothelial cells of CpKO mice (Appendix Fig S4).

## Cp administration prevents the loss of Purkinje cells in CpKO mice

Previous studies on CpKO mice have reported loss of dopaminergic and retinal neuron between 18 and 24 months of age (Patel *et al*, 2002; Jeong & David, 2006), and loss of cerebellar neurons as early as 12 months (Patel *et al*, 2002; Jeong & David, 2006). We performed Purkinje neuron cell counts on brain section stained with Toluidine blue. CpKO mice 10 months of age showed a significant loss of Purkinje neurons compared to WT mice, and treatment with Cp promoted a rescue of Purkinje cell loss (Fig 7A and B). This suggested that amelioration of motor coordination observed after Cp treatment might be the consequence of a reduction in cerebellar neuron degeneration.

## Unsupervised multivariate analysis shows that CpKO mice treated with Cp display a brain-related pattern similar to WT animals

Analysis was performed considering together the seven parameters related to brain that showed differences between CpKO and WT mice (Appendix Table S1). Two different dimensional reduction methods, namely principal component analysis and minimum curvilinearity embedding (linear and nonlinear unsupervised machine learning algorithms, respectively), confirmed the same result: CpKO mice treated with Cp (rose circular spots in Fig 8) collocated together with the WT mice groups, both Cp-treated and untreated (gray and black circular spots in Fig 8, respectively), and were clearly distinct from the untreated CpKO mice (red triangles in Fig 8). This distinction was underlined by the results of the unsupervised clustering analysis. This finding provides a clear evidence that supports the therapeutic efficacy of the Cp treatment in aceruloplasminemia.

**Figure 5.   Evaluation of anti-Cp immunological response against administered purified human Cp in mice serum.**

A   Detection by ELISA of anti-Cp IgG response in mice sera. Bckg, background; anti-mouse and anti-sheep, secondary antibodies; anti-Cp, commercial Ab used at different dilutions as positive control.

B   ELISA analysis of the ability to bind Cp of the IgG fraction purified from mice sera.

C   Ferroxidase activity analysis, measured with bathophenanthroline assay, on human-purified Cp incubated with IgG anti-Cp both as a mixture of commercial antibodies (anti-Cp) and IgG purified from mice. Cp heat inactivated (100°C) was used as protein inactivation control.

Data information: Data are presented as mean ± SEM of the animal groups average, and statistical significance was evaluated by Student's *t*-test. Each dot corresponds to one animal (CpKO, *n* = 5; CpKO + Cp, *n* = 6; WT, *n* = 5; WT + Cp, *n* = 6). Technical replicate *n* = 3 experiments.

## Discussion

In the present study, we show that enzyme replacement therapy (ERT) is a feasible therapeutic approach able to restore Cp ferroxidase activity in the brain of the preclinical mouse model of Acp. This mouse model has been reported to show neurological symptoms as consequence of the neurodegeneration induced by brain iron accumulation (Patel *et al*, 2002; Jeong & David, 2006). We focused our attention on the neurodegeneration features of the Acp for three main reasons: first, because the neurological symptoms are the more devastating pathology; second, because in humans, the onset of the neurological symptoms is preceded by about 10 years by the systemic symptoms (Miyajima, 2015a), which in turn provide an important therapeutic window exploitable for the prevention of brain iron accumulation and neurodegeneration (Vroegindeweij *et al*, 2015b); third, because current therapies are ineffective on neurodegeneration (Miyajima, 2015a).

Here we demonstrate that a 2-month administration of Cp by repeated intraperitoneal injections allowed Cp to reach the brain resulting in the replacement of enzymatically active protein. By single intraperitoneal injection, Cp was shown previously to be able to enter the brain of CpKO mice (Ayton *et al*, 2013); however, the kinetics of multiple injections, the protein accumulation, the rescue of ferroxidase activity, and, more relevant, the therapeutic efficacy on neurological symptoms have never been studied before. Differences in the age of onset of neurological symptoms have been reported in different CpKO mouse lines, which may depend on the genetic background of the mice and on the type of construct used for the generation of the Cp-null mice (Harris *et al*, 1999; Patel *et al*, 2002; Yamamoto *et al*, 2002; Jeong & David, 2006). Therefore, before starting the Cp treatment, we confirmed that 8-month-old CpKO mice already displayed deficit in motor coordination that revealed the onset of neurological

symptoms. This was in agreement with previous reports showing no behavior signs in CpKO mice at 3 months of age (Hineno *et al*, 2011; Texel *et al*, 2012) and the appearance of neurological symptoms at 5–6 months of age (Ayton *et al*, 2013, 2014). At the end of the Cp treatment, CpKO mice showed amelioration in motor coordination compared to untreated mice. Similar amelioration was seen in CpKO mice administered with the iron chelator deferiprone (Ayton *et al*, 2013). The improvement in motor coordination was associated with the presence of detectable, albeit variable, recovery of Cp levels in the brain of CpKO-treated mice. This indicates that the administered Cp accumulated in brain of CpKO mice to reach and maintain a paraphysiological level. On the contrary, a rapid Cp clearance from blood stream was observed. It has been reported that the half-life of circulating Cp is about 5.5 days *in vivo* (Hellman & Gitlin, 2002). Therefore, since Cp was almost undetectable in the serum of treated mice 5 days after the last injection, it is conceivable a reduced half-life of the administered Cp probably due to both protein sequestration in organs and faster degradation in serum. A quick serum degradation of exogenously administered Cp was also indirectly inferred from reported data showing a spike of serum iron concentration in CpKO mice soon after human Cp administration and a fast iron reduction after few hours (Harris *et al*, 1999). In spite of lower brain Cp levels in CpKO-treated mice compared to WT, we found a complete recovery of ferroxidase activity. This result can be explained considering that the administered Cp was purified from humans, which has been reported to be more active than the mouse Cp (Gray *et al*, 2009). The higher activity of human Cp might also explain why at the end of treatment, in the serum of CpKO mice, there was a small recovery of ferroxidase activity although the protein was almost undetectable.

In the Acp mouse model, brain iron deposition and neuronal loss have been described primarily in cerebellum and brainstem (Patel *et al*, 2002; Jeong & David, 2006; Ayton *et al*, 2013, 2014). We

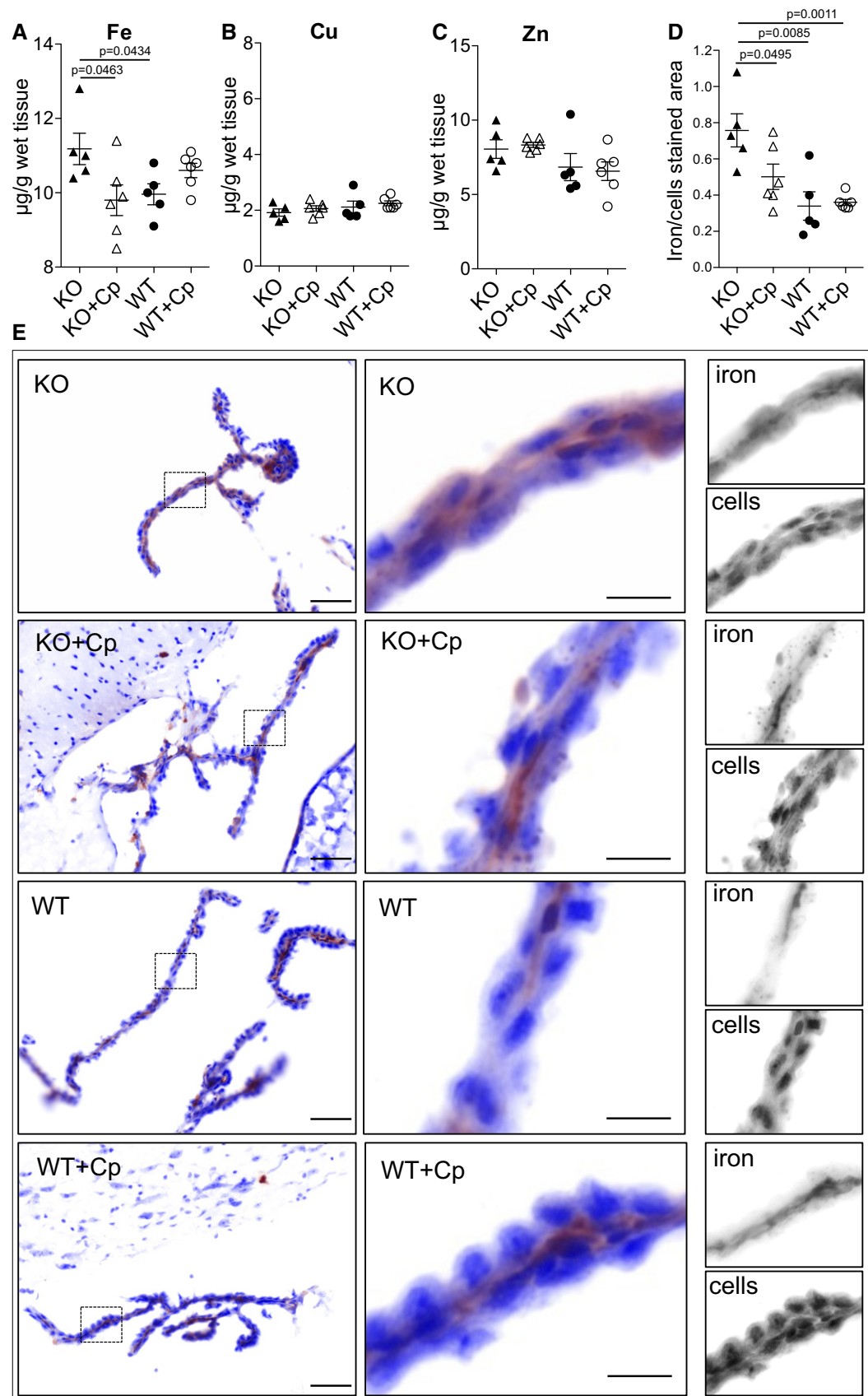

**Figure 6.**

**Figure 6.   Evaluation of brain iron accumulation.**

A–C   Metal ions quantification by inductively coupled plasma mass spectrometry in whole brain homogenates for iron (A), copper (B), and zinc (C). Data are presented as mean ± SEM (Student's *t*-test), and each dot corresponds to one animal (CpKO, *n* = 5; CpKO + Cp, *n* = 6; WT, *n* = 5; WT + Cp, *n* = 6).

D   Semi-quantitative analysis of the rate of iron/cell-stained areas of the images acquired at different length wave of emission (*n* = 10 fields for each mouse, technical replicate). Data are presented as mean ± SEM (Student's *t*-test), and each dot corresponds to one animal (CpKO, *n* = 5; CpKO + Cp, *n* = 6; WT, *n* = 5; WT + Cp, *n* = 6).

E   Iron staining (in brown) in choroid plexus epithelium, cells are counterstained in blue. Images acquired with Nuance® FX multiplex image system (PerkinElmer) at different wavelengths were computationally reassembled in pseudo color. Enlargement of the area indicated in left panel insertions and their corresponding gray-scale images originally acquired for different wavelengths are also shown. Scale bars = 50 and 10 μm for low and high magnification, respectively.

found whole brain iron accumulation in CpKO mice that fits with the reported iron increase detected in substantia nigra and cerebellum of 5- and 12-month-old CpKO mice (Jeong & David, 2006; Ayton *et al*, 2013, 2014). Interestingly, such iron accumulation was reduced by Cp treatment underlining its possible therapeutic use. The Cp treatment was specifically effective for iron, as the amount of copper and zinc ions was not affected. This is in agreement with the observations that no change in copper level has been found in the substantia nigra of CpKO mice (Meyer *et al*, 2001; Ayton *et al*, 2014). The copper content in the brain of Acp patients has been evaluated in few cases and was found not altered (Morita *et al*, 1995; Yoshida *et al*, 1995). However, a recent report showed a copper accumulation in the iron-rich particles in the brain of three Acp patients (Yoshida *et al*, 2017), and a copper level elevation in two patients was mentioned in Miyajima (2015b). On the contrary, a tendency to decrease in zinc concentrations in the brain of two Acp patients has been reported (Miyajima, 2015b).

Histochemistry analysis did not reveal iron deposition in brain of 10-month-old CpKO mice at the level of cerebellum and brainstem, but strong iron staining was observed at the level of choroid plexus epithelium. Previous findings also showed no obvious iron staining in the brain of 12-month-old CpKO mice, with the exception of weak signals in the ependymal cells lining the fourth ventricle (Jeong & David, 2006). In fact, iron staining in cerebellum and brainstem regions has been reported to become detectable in CpKO mice by 24 months (Patel *et al*, 2002; Jeong & David, 2006). Increased

intracellular iron in choroid plexus epithelial cells of CpKO mice was also inferred by the reported increase in ferritin expression in 24-month-old mice (Rouault *et al*, 2009). Thus, our results indicate the choroid plexus as the first brain region where, in the absence of Cp expression, a strong iron dis-metabolism occurs fostering iron accumulation. It is important to note that choroid plexus epithelial cells, that primarily constitute the blood–CSF barrier (B-CSF-B), normally do express Cp and thus can be profoundly affected by its absence in Acp (Klomp *et al*, 1996; Patel *et al*, 2000; Rouault *et al*, 2009; Marques *et al*, 2011). Remarkably, the iron deposition we found in choroid plexus of CpKO mice was reduced by Cp administration, supporting the efficacy of the ERT.

From the ERT point of view, an important issue is how Cp enters the CNS. The observation that the administered Cp did not apparently enter the brain in WT mice leads us to hypothesize the existence of a leakiness of brain barriers in CpKO mice. The iron accumulation found in choroid plexus might be responsible or contribute to the leakiness of the B-CSF-B. In fact, morphometric and functional changes of the choroid plexus epithelia have been reported in Alzheimer's disease as consequence of iron homeostasis alterations and oxidative stress induction, which in turn lead to B-CSF-B dysfunction (Perez-Gracia *et al*, 2009; Krzyzanowska & Carro, 2012; Mesquita *et al*, 2012; Serot *et al*, 2012). However, there are no reports of iron accumulation in the choroid plexus in Acp patients, likely due to little attention paid to this brain region. On the other hand, iron deposition was shown in perivascular astrocytes

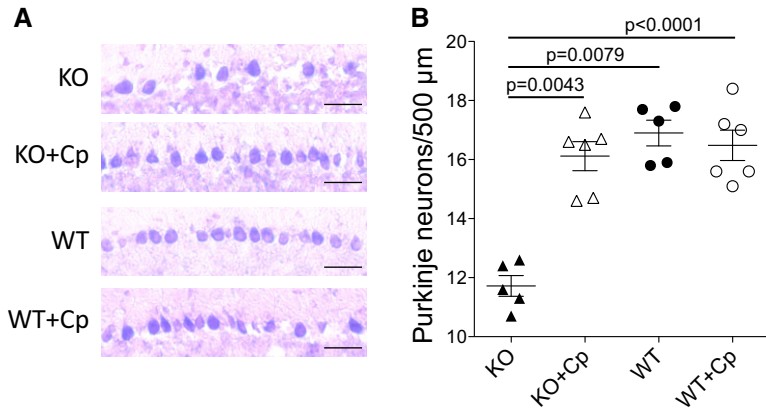

**Figure 7.   Purkinje cells count.**

A   Representative staining of Purkinje cells layer in the cerebellum of different mouse groups (scale bars = 50 μm).

B   Evaluation of Purkinje cells quantified on a linear distance of 500 μm in the Purkinje layer of cerebellum. Twenty linear distances were counted for each mouse (technical replicate *n* = 20). Data are presented as mean ± SEM (Student's *t*-test). Each dot corresponds to one animal (CpKO, *n* = 5; CpKO + Cp, *n* = 6; WT, *n* = 5; WT + Cp, *n* = 6).

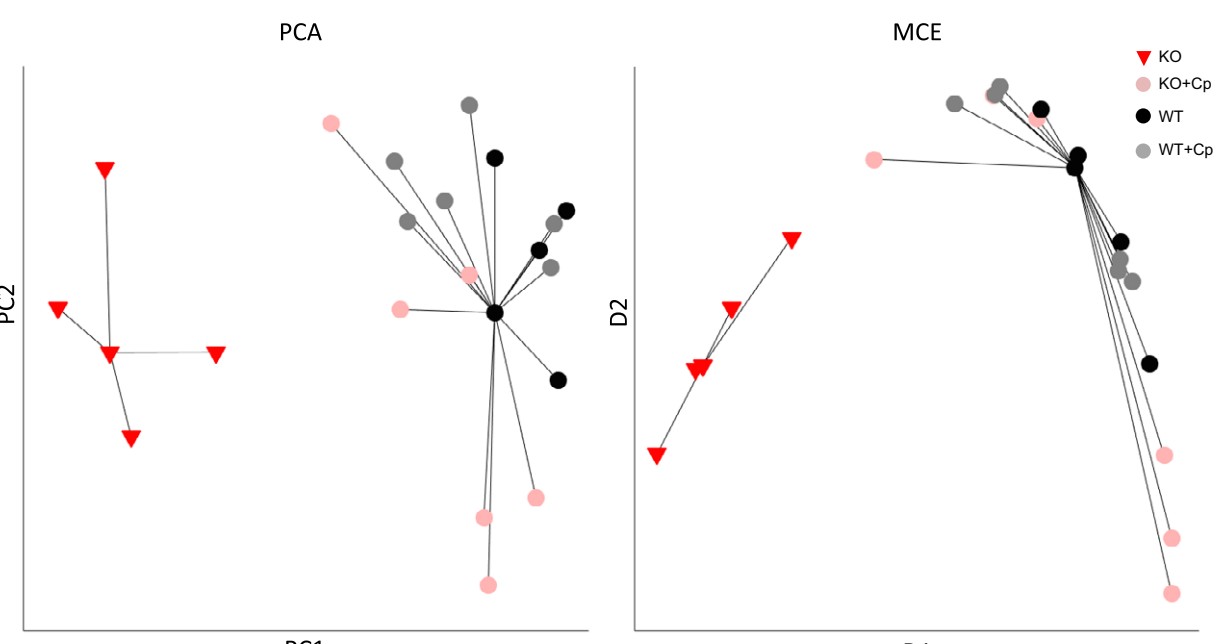

**Figure 8. Unsupervised multivariate analysis of the brain-related features.**

Unsupervised principal component analysis (PCA) and minimum curvilinear embedding (MCE) algorithms for dimensionality reduction were used. Each dot represents one animal. Unsupervised minimum curvilinear affinity propagation clustering analysis was applied in the two-dimensional reduced space of PCA and MCE. Each cluster is represented by black lines that connect the centroid, which is the sample representative for the cluster, with the other members of the cluster.

surrounding endothelial cells of brain microvasculature, suggesting a possible dysregulation of iron homeostasis in these structures (Kaneko *et al*, 2002; Oide *et al*, 2006; Gonzalez-Cuyar *et al*, 2008). We did not detected iron deposition, similar to that observed in choroid plexus, in the endothelial cells of the CpKO mice brain microvasculature, the primary elements of the blood–brain barrier (BBB). However, Cp expression and function in endothelial cells are not well defined (Klomp *et al*, 1996; Burkhart *et al*, 2016). Thus, it could be that BBB is less impaired than B-CSF-B, at least in the early stage of Acp neurological damage. It has been reported that Cp can pass from blood to hepatic parenchyma at the level of sinusoidal endothelial cells by transcytosis during which Cp is desialylated (Tavassoli, 1985; Irie & Tavassoli, 1986; Tavassoli *et al*, 1986). Nevertheless, our results did not support endothelial transcytosis as a prevalent mechanism of exogenous Cp entry into the brain, at least in the early stage of the neurological phase of the disease. Further studies on *in vivo* Cp biodistribution (e.g., by PET analysis of [64]Cu-labeled Cp administration) and *in vitro* in CpKO-cellular model of brain-barrier systems will be useful for the investigation of the hypothesized barriers permeability and leakage mechanism.

In CpKO mice, the reduction in Purkinje cells in cerebellum has been reported in 12- to 24-month-old CpKO animals (Patel *et al*, 2002; Jeong & David, 2006). Similar loss of Purkinje cells was clear also in CpKO mice in our present study, and the Cp replacement was efficacious in promoting the survival of these cells. The prevention of Purkinje cells loss in CpKO-treated mice might explain the amelioration observed in the motor coordination behavioral tests. Interestingly, a marked loss of Purkinje cells has been reported in human Acp, where the cerebellum appeared slightly shrunken and patients developed cerebellar ataxia (Morita *et al*,

1995; Miyajima *et al*, 2001; Miyajima, 2003; Kono, 2012). Two meta-analysis of 55 and 71 cases described worldwide up to the year 2015 showed cerebellar ataxia in the 41.5 and 48.2% of patients, respectively (Miyajima, 1993–2017; Vroegindeweij *et al*, 2015b). These reports indicated that in about 50% of Acp patients, the cerebellar phenotype may be similar to that of the preclinical model suggesting that Cp replacement therapy might be efficacious also in humans. It has been suggested that, both in Acp preclinical model and patients, neuronal loss appears firstly likely due to the iron depletion to neurons resulting from iron sequestration in astrocytes, which are the cells that primarily express Cp (Miyajima *et al*, 1996; Yoshida *et al*, 2000; Jeong & David, 2006; Kono, 2012; Miyajima, 2015a). Once dead, astrocytes release iron in the environment, and this might contribute to neuronal oxidative damage exacerbating their loss (Jeong & David, 2006). We can hypothesize that, at 10 months of age, Purkinje cells are dying likely by iron depletion, which is not yet deposited in the cerebellum, but rather is sequestered in the choroid plexus. Thus, the oxidative damage consequent to iron accumulation, reported as lipid and protein oxidation in older mice (Patel *et al*, 2002), is not yet occurring, as inferred by the similar level of protein carbonylation observed in CpKO and WT mice. Hineno *et al* (2011) also showed a lack of protein and lipid peroxidation in CpKO mice of 14 months. The indication that oxidative stress is not yet occurring in these mice could be an advantage from the therapeutic point of view. Indeed, the pro-oxidant environment is able to induce Cp modifications that result in a loss of ferroxidase activity (Olivieri *et al*, 2011; Barbariga *et al*, 2014, 2015) that might nullify the efficaciousness of the ERT. Another issue that might affect the efficacy of the ERT is the possibility that the administration of an exogenous protein

could induce a serological immune response. In our study, despite the high degree of homology between human and mouse Cp (> 80%), both CpKO and WT mice injected with human Cp developed an immunological response by generating anti-Cp antibodies. Nevertheless, the immunoglobulins from sensitized animals did not show neutralizing effect on human-purified Cp enzymatic activity. However, we cannot exclude the possibility that the trend in reduction in ferroxidase activity observed in Cp-treated WT mice might be induced by anti-Cp IgGs that resulted in specifically neutralizing mouse Cp and that might cross the brain-barrier systems by a regulated mechanism. This issue could be nullified or reduced by using species-specific protein, but purified mouse native Cp or functionally active mouse recombinant Cp was not available, and thus, human-purified native Cp was used in this study. However, it must be considered that also the administration of species-specific Cp cannot exclude "*a priori*" the eliciting of an immunological response against a protein that is not expressed by the receiving organism.

In brain, the Cp isoform prevalently expressed is the GPI-anchored Cp, compared to the secreted one (Patel *et al*, 2000); whether these two isoforms have distinct functional roles in brain is not yet clarified. Increasing evidence indicates that many cells that do express GPI-Cp, like astrocytes, do also express the secreted isoform (Greco *et al*, 2010; McCarthy & Kosman, 2014, 2015). Furthermore, several GPI-anchored proteins are controlled in their functions by a tightly regulated release from the membrane through the phospholipase-C enzyme, a mechanism that could be used to interconvert membrane to soluble Cp function (Patel & David, 1997). Since the administered Cp is the soluble form, if the two Cp isoforms have distinct functional roles, there is the possibility that ERT did not properly or fully act. However, we demonstrated that administered Cp was able to ameliorate the disease symptoms, indicating that the two isoforms may contribute to the same functions. Indeed, it has been already documented that soluble Cp is able to regulate iron homeostasis in neurons and brain microvascular endothelial cells (Ke *et al*, 2005; Olivieri *et al*, 2011; McCarthy & Kosman, 2013, 2014). Therefore, even if the replacement with soluble Cp could not reproduce exactly the physiological conditions supported by the presence of both GPI-anchored and soluble isoforms, from the therapeutic point of view, the amelioration of neurological symptoms observed is of prime importance. That ERT was efficacious in reducing neurodegenerative symptoms in the Acp preclinical model was further highlighted by the findings of the unsupervised multivariate analysis that confirmed the same result (efficacy of Cp treatment) using two different algorithms for dimensionality reduction, namely PCA and MCE. The latter in particular is suitable for small data sets in which, like in our study, the limited number of subjects is characterized by several features (Cannistraci *et al*, 2010; Alessio & Cannistraci, 2016). Considering together the different brain-related features characterizing the state of each mouse, CpKO-treated mice clearly collocated with WT rather than CpKO mice, suggesting that treated animals share a similar multivariate brain-related pattern with WT mice.

In conclusion, the results obtained indicate that the reconstitution of the Cp ferroxidase activity in the CNS is important, and feasible, in order to restrain neurodegeneration in the Acp preclinical model. We did not investigate in detail the effect of the Cp

ERT on the systemic deficiencies in CpKO mice, but we found that the significant high iron concentration detected in the liver of CpKO mice showed a trend in reduction after Cp treatment. The lack of a full effect might be due to the amount of iron detected in the liver of CpKO mice that was much greater than iron detected in brain; thus, the amount of the administered Cp reaching the liver is probably not enough to efficiently reduce iron accumulation. Longer Cp treatment or increased dose of administered Cp might lead to full recovery of physiological conditions also in liver. Similarly, Harris *et al* (1999) reported that Cp administration can mobilize iron out of the liver and temporarily restore iron homeostasis also in blood of CpKO mice. Moreover, a transient rescue of serum iron mobilized from stores has also been shown in Acp patients after administration of FFP containing ceruloplasmin (Logan *et al*, 1994; Yonekawa *et al*, 1999). Thus, the proposed ERT might be also efficacious on Acp systemic symptoms in addition to the neurological ones. An effective therapy for Acp is not available currently; however, based on the present study, Cp ERT might be considered a therapeutic opportunity also in humans. This can be examined even if the leakiness of B-CSF-B system would not be exploitable in human Acp, due to the lack of evidence supporting iron accumulation in choroid plexus of patients. Indeed, it would be worth to explore alternative systems for Cp delivery to the CNS or an intraventricular Cp expression (e.g., engineered Cp suitable for receptor-mediated transcytosis, nanoparticles, ependymal cells directed gene therapy). In particular, due to the late onset of neurological symptoms, it would be of interest to investigate in the future the effect of an early ERT in the prevention of neurodegeneration. Since in few cases, the iron chelators therapy combined with FFP administration appeared to be beneficial (Yonekawa *et al*, 1999; Poli *et al*, 2017), and since the serum Cp concentration has been reported to range between 21 and 54 mg/dl (Gibbs & Walshe, 1979; Miyajima, 1993–2017; Gaasch *et al*, 2007), it is conceivable that high-Cp-content FFP transfusions, instead of using random FFP, would be more effective. In fact, to reach a post-transfusion, Cp level of 8–10 mg/dl, which is the level that occurs in Acp heterozygotes, should be enough to rescue iron homeostasis as demonstrated by the absence of clinical symptoms in heterozygotes (Miyajima, 1993–2017; Kono, 2012). This would be a provisional para-ERT waiting for further experimental confirmation of our results on Cp ERT.

## Materials and Methods

### Animals used in the study

Ceruloplasmin-knockout mice on a C57Bl/6J genetic background from the original strain were used (Patel *et al*, 2002; Jeong & David, 2006). Since no documented differences in Acp penetrance or features have been reported, both male and female were used (Harris *et al*, 1998; Patel *et al*, 2002; Jeong & David, 2006). Age- and sex-matched wild-type C57Bl/6J mice (Charles River) were used as control population. Both WT and CpKO mice were randomly assigned to each experimental group at the begin of the study. Sample size was chosen according to previous experience with similar experimental models and assays. Investigators were not blinded during the experimental analysis. The study was approved by the

Institutional Animal Care and Use Committee (IACUC ID 687, San Raffaele Hospital) and by the National Ministry of Health (MoH no. 763/2015-PR).

### Treatment with Cp

In pharmacokinetics experiments, CpKO mice (3 months of age) were treated with human-purified Cp (Alexis Biochemicals) by intraperitoneal injection (5 μg/g in saline; two injections, at day 0 and at day 5). Three CpKO mice each time point (day 0, 1, 2, 5, 7, and 10) were euthanized and analyzed for Cp levels and ferroxidase activity. For therapeutic treatment, CpKO mice (*n* = 12) were aged till 8 months of age when early neurological symptoms appeared, assessed by motor coordination tests. CpKO and WT mice groups (*n* = 6 each) were treated for 2 months with purified Cp (5 μg/g in saline) administered intraperitoneal every 5 days (total of 12 injections), or with saline alone as control groups, both CpKO and WT (*n* = 6 each). Males and females were distributed to the different groups so as to match both the CpKO *vs.* WT populations and the Cp-treated *vs.* Cp-untreated groups. At the end of the treatment, after the behavior tests, the mice were bled from the retro-orbital plexus and then were euthanized by transcardial perfusion with ice-cold saline under deep anesthesia and brains were collected.

### Behavioral tests

The presence of neurological damage was evaluated as skill in motor coordination at the begin and at the end of the Cp treatment. Mice were tested for accelerating speed Rotarod assay and were submitted to five trials with an interval of 30 min. The mice were placed on the apparatus with the rod rotating at 4 rpm during the first minute, and then, the rotation speed was increased every 30 s by 4 rpm. A trial ended for a mouse when it fell down or when 5 min was completed. The latency to fall off the Rotarod was taken as the dependent variable for each trial. Motor coordination was also evaluated by the grid test in which the mouse was placed on a grid (1 × 1 cm mesh) and left free to walk for 2 min under video camera recording. The total number of steps and the number of times in which the animal's paws fall through the mesh of the grid were detected. The test was repeated three times, and coordination was evaluated as number of failure normalized to the total number of steps.

### Evaluation of Cp protein levels by Western blot analysis

The left brain hemisphere was homogenized in the presence of lysis buffer (20 mM Tris, 150 mM NaCl, 1% Triton X-100, protease inhibitors); protein from brain extracts and sera was resolved on 10% acrylamide SDS–PAGE and transferred onto a nitrocellulose membrane for Western blot analysis as reported in Conti *et al* (2005). The antibodies used were as follows: sheep anti-Cp (Abcam, ab8813), mouse anti-beta actin (Sigma, A5441), and appropriate secondary HRP-conjugated antibodies (working dilution 1:1,000). Signals were detected using ECL™ reagent (GE-Healthcare) followed by films exposure and densitometric analysis performed using ImageJ software (Rasband, W.S., ImageJ, U. S. National Institutes of Health, Bethesda, MD, USA). Signals were normalized to the total protein loaded and to actin expression (Conti *et al*, 2008).

### Analysis of sialylation status of Cp

The sialylation analysis was performed with the DIG-Glycan Differentiation Kit (Roche, 11 210 238 001). Brain homogenates and human-purified Cp were resolved by SDS–PAGE and analyzed by Western blot using lectins *Sambucus nigra* agglutinin (SNA) and *Maackia amurensis* agglutinin (MAA), which specifically recognize sialic acid. Membranes were incubated with lectins conjugated with digoxigenin (DIG; SNA 0.5 μg/ml, MAA 2.5 μg/ml), and reactivity was revealed by incubation with HRP-conjugated rabbit anti-DIG antibody (Dako, P5104; working dilution 1:1,000) followed by ECL reaction and films exposure. In selected experiments, in order to remove sialic acids, samples were incubated (18 h at 37°C) with 40 mU of *Vibrio cholerae* neuraminidase (Roche Diagnostics) in 50 mM sodium acetate (pH 5.5).

### Evaluation of ferroxidase activity

Ceruloplasmin ferroxidase activity was evaluated using the apo-transferrin assay in brain (Ayton *et al*, 2013) and bathophenanthroline assay (Huberman & Perez, 2002; Grundy *et al*, 2004; Olivieri *et al*, 2011) in serum, so as to better fit with the specific tissue background due, for example, in the sera, to the presence of large amount of transferrin in unpredictable apo- or holo status.

Bathophenanthroline specifically forms a complex with iron in its ferrous form leading to a red compound; when iron is oxidized it is released resulting in reduced absorbance. Sera (100 μg of total protein) were incubated (1 h at 37°C) with 85 μM ferrous sulfate (FeSO$_4$) in 0.2 M acetate buffer, pH 6.2; then, 1 mM bathophenanthroline was added and mixture incubated for 5 min at 20°C. Absorbance was measured in quadruplicate at 490 nm with microplate reader. Bathophenanthroline incubation with 85 μM ferric ammonium citrate was done for the estimation of the absorbance when the total iron present in the assay was fully oxidized. In the apo-transferrin assay, ferrous iron when oxidized by Cp to the ferric form is loaded into apo-transferrin causing a color change with absorbance at 460 nm. Brain homogenates (30 μg protein) were incubated (5 min at 37°C) with 74 mM acetate buffer pH 7.2, 55 μM apo-transferrin (BBI™ Group, T100-5), and 110 μM ammonium ferrous sulfate. Absorbance was then measured in quadruplicate at 460 nm. Since Cp activity is selectively inhibited by sodium azide (Gray *et al*, 2009; Ayton *et al*, 2013), 1 mM NaN$_3$ was used in order to ensure the specificity of Cp ferroxidase activity. After subtraction of sample blank to exclude the spontaneous background oxidation of ferrous iron, the ferroxidase activity was calculated by subtracting the NaN$_3$-inhibited value from the original sample value and was reported as percentages of the average of the activity evaluated in the WT mice. In selected experiments, the absolute amount of oxidized ferrous iron in different samples was reported considering the absorbance of the total amount of ferrous iron present in the assay (21.2 μg). In these experiments, sample ferroxidase activity was compared to the activity of human-purified Cp (1 μg).

### Detection of anti-Cp antibodies in sera and analysis of their neutralizing effect on Cp ferroxidase activity

ELISA plates were coated with purified human Cp (50 ng/well), 16 h at 4°C. Then, wells were incubated with blocking solution

(3% BSA in PBS) followed by incubation (2 h at 20°C) with mouse sera (1:150 dilution, corresponding to about 50 µg/µl total protein) in triplicate; sheep anti-Cp (Abcam, ab8813) antibody at different dilution or 1% BSA in PBS was used as positive and negative controls, respectively. The plates were then washed and incubated (1 h at 20°C) with HRP-conjugated secondary antibodies (anti-mouse IgG, 1:1,000, and anti-sheep IgG, 1:5,000). After washing, the assay was developed using o-phenylenediamine (Sigma) and 0.05% hydrogen peroxide, the reaction was stopped by adding 10% sulfuric acid, and absorbance was measured at 490 nm.

IgG fraction from mouse sera was purified with rProtein-G-agarose beads (Invitrogen). Sera (750 µg of total protein) were incubated (24 h at 4°C) with protein-G-agarose under gently stirring. Beads were washed; IgG was eluted with 0.1 M glycine, pH 2.6, and quickly buffered with 1 M Tris–HCl, pH 10 (1/10 v/v). The retention of the ability to bind human Cp by the purified IgG was tested by ELISA as described above, using 150 ng of purified IgG from each mouse serum against 50 ng of Cp. Neutralizing effect of IgG purified from mice sera on the Cp ferroxidase activity was analyzed by bathophenanthroline assay. Purified human Cp (600 ng) was incubated with 3 µg of purified IgG from each mouse serum, and bathophenanthroline assay was performed as described above. Cp alone or heat-inactivated Cp (5-min incubation at 100°C), or incubation with 3 µg of a mixture of anti-Cp commercial antibodies in equal amount (Abcam: ab8813, ab48614, ab19171; SantaCruz: sc21242, sc 21240) was used as control conditions.

## Analysis of metal ions by inductively coupled plasma mass spectrometry (ICP-MS)

Iron, copper, and zinc concentrations in the brain homogenates from microdissected organs were evaluated after digestion (1 h at 70°C) in a mixture of 65% nitric acid and 30% hydrogen peroxide. The metal content in the samples was determined by ICP-MS using an ELAN DRC II instrument (Perkin Elmer Sciex). The total quant technique analytical method, with external calibration using a dynamic reaction cell, was adopted. The coefficients of variation ranged from 4.5 to 7.6% among analytical series and from 5 to 10.5% between the series. The instrument was calibrated using standard solution at a concentration of 10 µg/l (Multielement ICP-MS Calibration Standard 3, Perkin Elmer Plus). Each sample underwent twofold determination. The accuracy of the method was calculated in ultrapure water. Bovine liver standard reference materials (NIST 1640 and MS1577b, National Institute of Standard and Technology, Gaithersburg, MD, USA) were used to better approximate the results from biological matrices. The detection limit, determined on the basis of three standard deviations of the background signal, was determined at 0.005 µg.

## Iron histochemistry

Modified Perl's histochemistry was performed on 14-µm cryostat sagittal sections of 4% paraformaldehyde-fixed brain. After a preliminary incubation with 2% ammonium sulfide, sections were incubated with 10% potassium ferricyanide and 0.5% HCl, followed by incubations with 0.025% diaminobenzidine and 0.005% $H_2O_2$, and counterstained with 0.5% Cresyl violet. Staining at different emission wavelength was evaluated by the Nuance® FX multiplex biomarker imaging system (PerkinElmer) in 10 different fields for

## The paper explained

### Problem

Aceruloplasminemia is a rare genetic disease caused by mutations in the ceruloplasmin (Cp) gene that result in protein loss or in a protein that has defective ferroxidase activity. Since Cp plays a role in iron homeostasis, the absence of its activity leads to iron accumulation in liver, pancreas, and brain. Iron deposition promotes diabetes, retinal degeneration, and the development of a severe progressive neurological degeneration. Current therapies, mainly based on iron chelation, partially control systemic iron deposition but are ineffective on neurodegeneration and showed side effects so that they must be discontinued. Therefore, there is a need for therapeutic strategies effective on neurological degeneration.

### Results

We show the potential of Cp replacement therapy in reducing the neurological pathology in the Cp-knockout (CpKO) mouse model of aceruloplasminemia. CpKO mice, displaying neurological symptoms, were treated with Cp for 2 months. The intraperitoneal-administered Cp was able to cross the brain-barrier systems and to enter the brain inducing both a replacement of the protein levels and a rescue of ferroxidase activity. As sign of neurological improvement, Cp-treated mice showed amelioration of motor incoordination that was associated with diminished loss of Purkinje neurons and reduced brain iron deposition. Biocomputational analysis showed that Cp-treated CpKO mice cluster with the wild-type animals highlighting the efficacy of the Cp replacement therapy in this preclinical model of aceruloplasminemia.

### Impact

Our study suggests that enzyme replacement therapy may be a promising strategy for aceruloplasminemia treatment also in humans. The Cp replacement therapy would provide a treatment effective on the devastating neurological degeneration that is responsible for the long-lasting neurological disabling period of life of the patients. Furthermore, the proposed therapy could be transposed in short term to human clinical trial because Cp can be easily purified from human plasma as injectable derivate.

each mice. A semi-quantitative analysis of iron deposition normalized for cell staining in areas of defined size was performed using ImageJ software.

## Purkinje cells count

Purkinje cell histology was performed using 0.1% Toluidine blue staining on 14-µm cryostat sections of 4% paraformaldehyde-fixed brain. Sections were then mounted with coverslip using Mowiol® 4-88. Samples were analyzed with a Zeiss AxioImager M2m microscope, and images acquired with AxioCam MRc5 (Zeiss). Images were analyzed with ImageJ software for automated count of Purkinje body cell over a linear distance of 500 µm in the Purkinje cell layer of the cerebellum. Twenty different linear distances were analyzed for each mouse.

## Statistical analyses

The data from the different groups of animals were evaluated by unpaired Student's *t*-test, if the data passed the normality test for Gaussian distribution (Kolmogorov–Smirnov test), or were

evaluated by Mann–Whitney *U*-test; two-tailed *P*-value was used for the comparison of two means and standard error. In all analyses, $P < 0.05$ was considered to be statistically significant. The analyses were performed with Prism V5.0 software (GraphPad Inc.).

Multivariate analysis, which considers together the features collected for describing the brain pathophysiological state of each mouse, was done using unsupervised and parameter-free (the algorithms do not require the tuning of any internal parameters, and hence, they prevent bias and overfitting) machine learning algorithms for linear and nonlinear dimension reduction. We applied principal component analysis (PCA), which is a state-of-the-art approach for linear dimension reduction (Ringner, 2008; Ciucci *et al*, 2017) and minimum curvilinear embedding (MCE), which is a nonlinear and parameter-free kernel PCA whose efficacy was extensively tested in previous studies (Cannistraci *et al*, 2010, 2013; Ammirati *et al*, 2012; Alanis-Lobato *et al*, 2015; Alessio & Cannistraci, 2016; Sales *et al*, 2016). We adopted both PCA and MCE to have confirmation that the results displayed were substantiate regardless of the type of linear or nonlinear transformation employed for their analysis. In particular, we considered the features reported in Appendix Table S1. Since these features were measured according to different scales, they were normalized by *z*-score transformation to re-scale their values inside the same range and allow a comparable multivariate analysis. The presence of groups of affinity among animals was investigated by using the unsupervised Minimum Curvilinear affinity propagation clustering (MCAP) algorithm (Cannistraci *et al*, 2010, 2013) that was applied in the two-dimensional reduced space resulting from PCA and MCE analysis and was executed considering the option to detect two clusters. The analysis were performed using MATLAB v7.0 software.

**Expanded View** for this article is available online.

## Acknowledgements
This work was supported by Telethon Grant Exploratory Project (GEP-14102) with the exception of the open access publication costs. We thank the Advanced Light and Electron Microscopy BioImaging Center at San Raffaele Scientific Institute for imaging assistance.

## Author contributions
MA conceived and designed the study; SD provided the CpKO mouse model and discussed the experiments; AP contributed to ferroxidase analysis; AZ and MC maintained the mice colony and performed biochemical experiments; AC carried out histochemistry; PD'A performed behavioral tests; PA and EG performed metal ions analysis; CVC carried out multivariate analysis. All the authors discussed the results and commented on the manuscript at all stages.

## Conflict of interest
The authors declare that they have no conflict of interest.

## For more information
OMIM, Ceruloplasmin gene: https://www.omim.org/entry/117700?search=ceruloplasmin%20&highlight=ceruloplasmin
OMIM, aceruloplasminemia: https://www.omim.org/entry/604290
NCBI, Ceruloplasmin gene: https://www.ncbi.nlm.nih.gov/gene/1356
UniProtKB, human Ceruloplasmin: http://www.uniprot.org/uniprot/P00450
UniProtKB, mouse Ceruloplasmin: http://www.uniprot.org/uniprot/Q61147

NCBI Books, Aceruloplasminemia: https://www.ncbi.nlm.nih.gov/books/NBK1493/
ORPHANET, Aceruloplasminemia: http://www.orpha.net/consor/cgi-bin/Disease_Search.php?lng=EN&data_id=10633&Disease_Disease_Search_disease
Group=aceruloplasminemia&Disease_Disease_Search_diseaseType=Pat&Disease(s)/group%20of%20diseases=Aceruloplasminemia&title=Aceruloplasminemia&search=Disease_Search_Simple
NBIA, Neurodegeneration with Brain Iron Accumulation Disorders Association: http://www.NBIAdisorders.org

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
