## [Review Process File · EMBO Molecular Medicine]

Ceruloplasmin replacement therapy ameliorates neurological symptoms in a preclinical model of aceruloplasminemia

Alan Zanardi, Antonio Conti, Marco Cremonesi, Patrizia D'Adamo, Enrica Gilberti, Pietro Apostoli, Carlo Vittorio Cannistraci, Alberto Piperno, Samuel David and Massimo Alessio

Corresponding author: Massimo Alessio, IRCCS-San Raffaele Scientific Institute

Review timeline:

Submission date:	04 August 2017
Editorial Decision:	29 August 2017
Revision received:	05 October 2017
Editorial Decision:	26 October 2017
Revision received:	26 October 2017
Accepted:	30 October 2017

Transaction Report:

Editor: Céline Carret

1st Editorial Decision

29 August 2017

Thank you for the submission of your manuscript to EMBO Molecular Medicine. We have now heard back from the three referees whom we asked to evaluate your manuscript.

You will see that while the referees find the study of significant interest they do have several suggestions that if followed, would strongly strengthen the study. It is our opinion that all suggested clarifications and text modifications would improve the impact of the paper and I would therefore encourage you to address these in a major revision of your work. Please note that it is EMBO Molecular Medicine policy to allow only a single round of revision and that, as acceptance or rejection of the manuscript will depend on another round of review, your responses should be as complete as possible.

I look forward to receiving your revised manuscript.

***** Reviewer's comments *****

Referee #1 (Comments on Novelty/Model System for Author):

This can be moved to clinical trial quickly.

Referee #1 (Remarks for Author):

Ceruloplasmin replacement therapy ameliorates neurological symptoms in a preclinical model of aceruloplasminemia

This is a very important paper. This reviewer has made many comments but they are relatively

minor and should be able to be dealt with easily by the authors. The figures do need a fair bit of work.

Abstract

Computational analysis showed that ceruloplasmin-treated CpKO mice share a similar pattern with wild-type animals, highlighting the efficacy of the therapy.

Remove. Computational analysis from paper

These data suggest that enzyme replacement therapy may be a promising strategy for aceruloplasminemia treatment

Change to

These data suggest that enzyme replacement therapy may be a promising strategy for the treatment of aceruloplasminemia

Introduction

"In humans with Acp, the absence of Cp ferroxidase activity leads" This needs to be expanded and qualified. E.g. What are the different types of ACP

"Multivariate dimensional reduction analysis showed that CpKO mice treated with Cp were similar to the WT animals, supporting the efficacy of Cp replacement therapy for Acp." This is not convincing and distracts from the excellent work in the paper.

Results

"Cp and administered human Cp was also evident in serum (Fig. 3A, arrows vs. black head-arrows). Apparently, none of the administered Cp was able to enter the brain of the WT animals, suggesting a difference in brain-barrier-systems permeability/leakage in CpKO mice " I think this takes the data too far and not convincing especially since the WT+ CP (fig 4) has altered brain ferroxidase activity. The variance in this group suggests that I it is getting into the brain and an increase in n would probable show a significant difference.

Discussion:

"This discrepancy might be explained by the different mouse lines and different methods used in evaluating dopaminergic neuronal loss."

Suggestion remove all TH data. "Brain" TH is not a valid assessment of neuronal number. Cpu or Sn th is used to assess loss of dopamine in motor control circuits but whole brain TH is a very poor measure. The original paper by Prof David phenotyping the mouse photographed the nigra and showed loss of cells. Ayton use stereological cell counts. The source of mice in the two studies appears to be the same.

Figures:

General comments.

1. The scatter plot with mean as deviation is preferred to the bar graphs. This provides more information about the data set. The mixture of bar and scatter not required and distracting.
2. The multiple bars to indicate significant is confusing. The number could be removed. Visually *, **, and *** would be simple.

Figure 1

1b title is incorrect. Relative Cp protein Level ?

Figure 1 figure ledgend "Cp expression" should be changed it is not expression as it is exogenous.

Figure 2 BCDE should be scatter graphs or remove. Why put in both accelerating and constant speed RR.

Figure 3 : A. The exogenous and endogenous images are not convincing, especially in the WT+ CP. The authors should consider removing this concept from the paper.

Figure 3 and 4. Why is there no apparent correlation between the amount of protein and ferroxidase activity across all samples. Please address.

Fig 4) Why is there less CP activity in brain following cp administration to the wt?

Figure 6d and E. Brown staining is not obvious. Brown stain is very difficult to see on a blue and pink background, Can sections be re stained? There are no scale bars. Remove the sentence images were taken at X40 . The Perls stain does not seem to have worked/

Figure 7. The loss of cells in the 10 month KO is spectacular, do you have images at 8 and 9 months.

Figure 8 Reviewer suggests Removing this figure it distracts and doesn't add to the argument.

Appendix

Change title "Evaluation of total brain proteins oxidation."

Please also address why there were no protein carbonyls in the KO group.

Delete the Appendix S3: The Perls stain does not seem to have worked. There is no scale bar on the figures. Remove sentence "Images were taken at x20". The WT has dirt in the image.

Delete appendix S4. "Brain" TH is not a valid assessment of neuronal number. Cpu or Sn th is used to assess loss of dopamine in motor control circuits but whole brain TH is a very poor measure. The original paper by Prof David phenotyping the mouse photographed the nigra and showed loss of cells.

Delete Multivariate analysis. S1

Referee #2 (Remarks for Author):

The paper by Zanardi et al reports the approach of enzyme replacement therapy in a knock-out mouse model for ceruloplasmin (CP), a protein playing key role in iron homeostasis. In human, mutations of CP cause aceruloplasminemia a disorder characterized by iron accumulation in liver, pancreas and brain causing diabetes, retinal degeneration and progressive neurodegeneration. The paper is potentially relevant despite several weaknesses are present and the data are not always clearly reported.

Specifically, a first observation is that up to the results session it is not evident that the protein used for ERT was of human origin. I think this crucial information should be reported in the abstract. As concerning the results I would like to comment on the following experiments:

1) In Figure 1 panel A there are only two animals /group while the treated animals were 3. In addition, only one untreated animal is present and no WT animals are reported just to make the comparison of the results easy to interpret.

Moreover, at day 5 the WB shows faint bands while the evaluation of Cp expression in panel B, is higher than that at day 1 or 2 where the WB shows an intense band. How the Authors explain these contradictory results?

2) In Figure 3 the WB in panel A is not very clear. I would suggest either to perform a less exposure to avoid saturation of signals or to run the gel for more time in order to obtain a better separation of the proteins and demonstrated the different MW of the human and mouse Cp protein. Is the Cp on the right the purified human protein? This should be specified in the legend of the figure. Moreover, the Authors claim that the externally administered CP was not able to cross the BBB in WT mice. Again this is very difficult to say from the presented picture.

Also in panel D, especially the right part, it is difficult to identify the band.

3) In Figure 4A it is reported the ferroxidase activity in the brain of mice. While it is clear the effect of Cp expression in KO animals, it is not clear why WT mice injected with Cp showed a reduced ferroxidase activity. This should be explained and motivated.

4) Despite the fact that neurodegeneration is the major cause of disability in human (and mice), the Authors should also consider iron accumulation in liver and pancreas as part of the disease spectrum. I would suggest to measure iron in liver of the Cp KO mice and to evaluate the effect of Cp administration. This is particularly important in evaluating the potential of ERT therapy in humans. In fact, Authors describe the possibility of a "preventive therapy" in a period in which neurological symptoms are not evident.

Which are the first signs of the disease in humans that could be monitored to start a preventive therapy?

5) The Authors describe loss of Purkinje cells as responsible for the movement disorder in mice. Is the same phenotype present in humans? Is cerebellar atrophy a common feature in aceruloplasminemia? This information should be added.

6) There is evidence for iron accumulation in choroid plexus epithelium in CpKO mice that is also responsible for the successful of the ERT with the human Cp protein, which can permeate thus reaching the brain.

However, in aceruloplasminemia iron accumulation is in other brain region and there are no evidence that the same phenomenon could guarantee the success of a therapy in humans. Could the Authors comment on this?

Referee #3 (Comments on Novelty/Model System for Author):

The manuscript was well written; the citation of the references is appropriate. Aceruloplasminemia is a slowly progressive neurodegenerative disorder, and an early diagnosis and early treatment of patients are important issues. To reduce iron accumulation, iron chelation therapy is effective for reducing the hepatic and pancreatic iron overload and improving diabetes, but not effective for neurological symptoms. In this paper, intraperitoneal-administered human ceruloplasmin was able to enter the brain inducing replacement of the protein levels and rescue of ferroxidase activity of ceruloplasmin-knock-out mice. Ceruloplasmin-treated mice showed amelioration of motor incoordination associated with diminished loss of Purkinje neurons and reduced brain iron deposition. These findings are very important for treatment of aceruloplasminemia patients

There are some issues I believe the authors need to address.

Combination therapy with fresh frozen plasma to replenish the blood ceruloplasmin levels, and thereafter the administration of desferrioxamine to deplete iron stores, showed an improvement in the neurological symptoms. However, after receiving long-term treatment with fresh frozen plasma by intravenous infusion once a week for two years, a brain MRI evaluation demonstrated improvement in the low intensity areas in the basal ganglia, but showed little effect on neurological symptoms. The ceruloplasmin replacement therapeutic approach remains controversial in patients with aceruloplasminemia. Iron deposition is more severe in the endothelial cells of the microvasculature than in choroid plexus in aceruloplasminemia brain. Distribution of iron deposition is different between aceruloplasminemia patients and ceruloplasmin-knock-out mice. It isn't necessarily true that ceruloplasmin replacement therapy ameliorate neurological symptoms in all patients with aceruloplasminemia.

Referee #3 (Remarks for Author):

Aceruloplasminemia is a slowly progressive neurodegenerative disorder, and an early diagnosis and early treatment of patients are important issues. To reduce iron accumulation, iron chelation therapy is effective for reducing the hepatic and pancreatic iron overload and improving diabetes, but not effective for neurological symptoms. In this paper, intraperitoneal-administered human ceruloplasmin was able to enter the brain inducing replacement of the protein levels and rescue of ferroxidase activity of ceruloplasmin-knock-out mice. Ceruloplasmin-treated mice showed amelioration of motor incoordination associated with diminished loss of Purkinje neurons and reduced brain iron deposition. These findings are very important for treatment of aceruloplasminemia patients

There are some issues I believe the authors need to address.

1. Combination therapy with fresh frozen plasma to replenish the blood ceruloplasmin levels, and thereafter the administration of desferrioxamine to deplete iron stores, showed an improvement in the neurological symptoms. However, after receiving long-term treatment with fresh frozen plasma by intravenous infusion once a week for two years, a brain MRI evaluation demonstrated improvement in the low intensity areas in the basal ganglia, but showed little effect on neurological

symptoms. The ceruloplasmin replacement therapeutic approach remains controversial in patients with aceruloplasminemia. Iron deposition is more severe in the endothelial cells of the microvasculature than in choroid plexus in aceruloplasminemia brain. Distribution of iron deposition is different between aceruloplasminemia patients and ceruloplasmin-knock-out mice. It isn't necessarily true that ceruloplasmin replacement therapy ameliorate neurological symptoms in all patients with aceruloplasminemia.

2. The authors described that the administered ceruloplasmin that reached the brain entered the central nervous system by a mechanism different than endothelial transcytosis. The authors should explain blood-brain barriers permeability and leakage mechanism in detail.

3. The authors described that no alterations in copper and zinc expression have been reported in aceruloplasminemia patients. In aceruloplasminemia brains, however, massive iron deposition was observed in the brain associated with low zinc content and high copper content.

1st Revision - authors' response

05 October 2017

Referee #1 (Comments on Novelty/Model System for Author):

This can be moved to clinical trial quickly.

Referee #1 (Remarks for Author):

Ceruloplasmin replacement therapy ameliorates neurological symptoms in a preclinical model of aceruloplasminemia

This is a very important paper. This reviewer has made many comments but they are relatively minor and should be able to be dealt with easily by the authors. The figures do need a fair bit of work.

Abstract

Computational analysis showed that ceruloplasmin-treated CpKO mice share a similar pattern with wild-type animals, highlighting the efficacy of the therapy.

The reviewer suggested to remove the Computational analysis from paper

Authors' answer: We thank the reviewer to offer us the opportunity to discuss this contradictory subject. We agree with the reviewer that computational methods based on machine learning and multivariate statistics can be sometimes more complicated than what the biological context requires. However, we would like to kindly point out that the usage of computational analysis was well-motivated in our case, and indeed it offered important points for discussion, especially to reject any data bias due to reduced sample size.

The point of strength is that these algorithms are not hypothesis-driven (as statistical tests) but data-driven, hence the analysis is unbiased by previous hypothesis on the origin or classification of the data. Furthermore, the adopted multivariate methods take into consideration the whole body of available information. This is important because in a living organism all the different components contribute to highlight the diverse physio/pathological conditions, regardless that, if considered as single component they are not significant *per se* in discriminating the different groups. Small coherent changes that are mutually associated can make large differences. The relevance of this kind of analysis for interpretation of the results of biological sample is underlined in the literature that are cited in the manuscript (Ringner, Nat Biotech 2008; Ciucci et al. Sci Rep 2017). We therefore feel that our computational analysis has added value to the manuscript. The original results of the multivariate analysis are not just a different statistical way to re-analyze the biological data, but they offer an important proof that the results are not affected by the sample size or previous hypothesis on the data, adding also complementary information on the general effect that Cp-ERT has on brain of KO mice. The figure related with the Multivariate analysis results

(fig. 8), shows clearly that KO+Cp samples have a multivariate profile that can be associated to the WT and WT+Cp. We think that this message is not confusing; because it stresses that the Cp treatment causes a collective and systemic effect that moves the KO+Cp mice towards a recovered condition.

In addition, since the reviewer complained about the clarity of the dimension reduction result, we performed clustering analysis that unsupervised detected the presence of two clusters among mice. The analysis was performed using the Minimum Curvilinear affinity propagation clustering algorithm (Cannistraci et al. *Bioinformatics* 2010 and 2013) that is based on the affinity propagation concept originally described in Frey and Dueck, *Science* 2007. The two resulting clusters, graphically represented in the new figure 8, clearly underline for the readers the similarity of the Cp-treated KO mice with the WT groups of animals. This new analysis has been introduced in the Materials and Methods section of the manuscript ("The presence of groups of affinity among animals was investigated by using the unsupervised Minimum Curvilinear affinity propagation clustering (MCAP) algorithm {Cannistraci, 2010 #542} {Cannistraci, 2013 #691} that was applied in the two-dimensional reduced space resulting from PCA and MCE analysis and was executed considering the option to detect two clusters."). Moreover, the following sentence has been added in the results section of the revised manuscript "This distinction was underlined by the results of the unsupervised clustering analysis.", and the figure legend has been accordingly modified.

Based on these reasons and in the light of the fact that the other two Referees did not issues with computational results, we would prefer to keep this part within the revised manuscript.

These data suggest that enzyme replacement therapy may be a promising strategy for aceruloplasminemia treatment

The reviewer suggested to change this to:

These data suggest that enzyme replacement therapy may be a promising strategy for the treatment of aceruloplasminemia

Authors' answer: done as requested.

Introduction

"In humans with Acp, the absence of Cp ferroxidase activity leads" The reviewer suggested that this needs to be expanded and qualified. E.g. What are the different types of ACP

Authors' answer: The description of Acp features has been expanded and text changed as follows: "Acp is caused by mutations in CP gene leading to absent or dysfunctional protein. So far, approximately 50 pathogenic variants of Cp have been identified world-wide in more than 60 family from different racial groups, even though the larger number of families have been identified in Japan (Kono, 2012; Miyajima, 2015). In humans with Acp, the absence of Cp ferroxidase activity leads, on one hand, to a decreased iron delivery to transferrin and iron-restricted erythropoiesis, and, on the other hand, to iron accumulation in the parenchyma of several tissues including liver, pancreas, heart, thyroid, retina and brain (Kono, 2012; Miyajima, 2015). Typical Acp manifestations are mild microcytic anemia and complications due to iron-mediated cytotoxicity. The latter, fostered by iron deposition, results in retinal degeneration, diabetes, hypothyroidism, cardiac failure that generally precede by about ten years the onset of neurological and psychiatric symptoms (Fasano et al, 2007; Kono, 2012; Miyajima, 2015; Vroegindeweyj et al, 2015b). The clinical manifestations at diagnosis are heterogeneous and range from a full-blown phenotype characterized by anemia, tissue iron overload and iron-related complications, to only iron-restricted anemia and tissue iron overload in the absence of non-hematological manifestations. For example, it has been reported that in 71 Acp patients only 68% displayed neurological signs, while 80%, 76% and 70% showed anemia, retinal degeneration and diabetes, respectively (Miyajima, 1993). Although different Cp mutations and genetic backgrounds might be implicated, data are not sufficient to draw conclusion explaining phenotype heterogeneity."

"Multivariate dimensional reduction analysis showed that CpKO mice treated with Cp were similar to the WT animals, supporting the efficacy of Cp replacement therapy for Acp." The reviewer stated that This is not convincing and distracts from the excellent work in the paper.

Authors' answer: see above.

Results

"Cp and administered human Cp was also evident in serum (Fig. 3A, arrows vs. black head-arrows). Apparently, none of the administered Cp was able to enter the brain of the WT animals, suggesting a difference in brain-barrier-systems permeability/leakage in CpKO mice " The reviewer stated that: I think this takes the data too far and not convincing especially since the WT+ CP (fig 4) has altered brain ferroxidase activity. The variance in this group suggests that it is getting into the brain and an increase in n would probably show a significant difference.

Authors' answer: We do not have a plausible explanation for this effect. The scenario can be various. On one hand, we should take into consideration, as this Reviewer claims, that some administered Cp is able to enter the brain also in WT mice. Although this should occur in undetectable amount, at least for the Western blot assay (less than 50 pg) because otherwise we should be able to detect it (see answer to Figure 3 comment). Then we should hypothesize that this tiny amount of exogenous Cp, instead of increasing the total ferroxidase activity, as expected, perturbs "in some way" the functionality of the physiological Cp, which in turn results in a paradoxical effect of reduction in ferroxidase activity not dependent on the endogenous Cp (i.e. expression was similar to the one observed in untreated WT mice). We are not in favor of this hypothesis because we are not able to detect administered Cp in the brain of WT mice. However, we agree with the Reviewer that the data do not directly "suggest" permeability/leakage in the barrier systems, but only a different ability of the administered Cp to enter the CNS of CpKO mice. Different permeability can only be speculatively hypothesized. Thus the sentence in the original results section, reported by the Reviewer, has been changed as follow: "Apparently, none of the administered Cp was able to enter the brain of the WT animals, at least in detectable amounts as in CpKO mice, suggesting a different ability of the administered Cp to enter the CNS in CpKO mice.". In addition, also in the discussion the sentence referring to the suggestion of barrier permeability has been change as follows: "The observation that the administered Cp did not apparently enter the brain in WT mice, lead us to hypothesise the existence of leakiness of the brain barrier in CpKO mice."

We can also speculate that some of the anti-Cp IgG found in the serum of WT mice, that did not showed a neutralizing effect on the *in vitro* ferroxidase activity of human Cp (see fig. 5C), might be inhibitory for the endogenous mouse Cp activity. In this case we should also take into consideration that IgGs might cross the brain barrier, in a regulated way and not by barriers leakiness, otherwise we should find also exogenous Cp in the brain of WT animals. A sentence reporting this concept has been introduced in the discussion section ("However, we cannot exclude the possibility that the trend in reduction of ferroxidase activity observed in Cp-treated WT mice might be induced by anti-Cp IgGs that resulted in specifically neutralizing the mouse Cp, and that might cross the brain-barrier y a regulated mechanism."):

In conclusion, even if it would be important to explain this paradoxical effect, if confirmed in larger set of animals. From the point of view of Acp treatment our results clearly demonstrated the potential therapeutic efficacy of ERT.

A sentence underlining this puzzling result has been added in the results section of the revised manuscript: "On the contrary, an unexpected non-significant trend in reduction of ferroxidase activity was observed in the brain of Cp-treated WT mice (Fig. 4A)."

Discussion:

"This discrepancy might be explained by the different mouse lines and different methods used in evaluating dopaminergic neuronal loss."

This Reviewer suggested to remove all TH data. The reviewer stated: "Brain" TH is not a valid assessment of neuronal number. Cpu or Sn th is used to assess loss of dopamine in motor control circuits but whole brain TH is a very poor measure. The original paper by Prof David phenotyping the mouse photographed the nigra and showed loss of cells. Ayton use stereological cell counts. The source of mice in the two studies appears to be the same.

Authors' answer: As suggested, the TH data presented as Appendix S4 have been removed from the revised version of the manuscript. Also the sentences present in the Results and Discussion sections that referred to these results were removed in the new version of the manuscript. We

removed this sentence from Results "~~By quantification of the expression of tyrosine hydroxylase (TH), a specific marker of dopaminergic neurons, we indirectly evaluated in the brain extracts the loss of these cells in 10-month-old mice. Western blot analysis, showed similar expression level of TH in all the animal groups, indicating that, at this stage of the pathology, the reduction in dopaminergic neurons was not yet detectable in CpKO mice (Appendix Fig. S4).~~", and from Discussion the sentence "~~In CpKO mice loss of dopaminergic neurons has been described in 6-24 months-old mice (Ayton et al, 2014; Ayton et al, 2013; Patel et al, 2002), but we did not find a reduction in tyrosine hydroxylase, a dopaminergic neuron marker, in 10 months-old CpKO mice. This discrepancy might be explained by the different mouse lines and different methods used in evaluating dopaminergic neuronal loss.~~".

Figures:

General comments.

1. The scatter plot with mean and s deviation is preferred to the bar graphs. This provides more information about the data set. The mixture of bar and scatter not required and distracting.

Authors' answer: scatter plot have been introduced in alternative to bar graphs for the experiment in which the data are referred to each single mouse, in order to have single dot as representative of a single animal. To avoid confusion, the bar graphs were maintained when data were referring to purified Cp activity (e.g. Fig.4C and 5C) or to control immunoglobulin reactivity in the ELISA test (e.g. Fig. 5A). Figure legends have been accordingly modified.

Also the statistical analysis have been *de novo* evaluated not considering the number of technical replicates in the groups but their averages in each mouse. As expected by introducing this change, the p value of statistical significance resulted to be modified in some analysis. In particular, in the behavioral test (new panel B and E in figure 2) and in the analysis of iron accumulation in choroid plexus (Fig. 6 new panel D) the p values between Cp-treated and untreated KO mice is lower but still significant. The results of these new analyses are in line with our previous conclusions that indicated an amelioration of the neurological symptoms fostered by Cp replacement, and underline the importance of the use multivariate analysis in the case datasets of reduced sample size (see answer above). The new p values are reported in the figure 2 and 6 of the revised manuscript.

2. The multiple bars to indicate significant is confusing. The number could be removed. Visually *, **, and *** would be simple.

Authors' answer: the number of statistical significance were indicated according to the EMBO-MM journal guideline request. "...and the actual P value for each test (not merely 'significant' or 'P<0.05')".

Figure 1

1b title is incorrect. Relative Cp protein Level ?

Authors' answer: we apologize for the mistake. The title has been changed as suggested.

Figure 1 figure legend "Cp expression" should be changed it is not expression as it is exogenous.

Authors' answer: "Cp expression" was not present in the Figure 1 figure legend, thus we thought that the Referee was referring to the title of panel 1b (see above).

Figure 2 BCDE should be scatter graphs or remove. Why put in both accelerating and constant speed RR.

Authors' answer: scatter plot graphs have been introduced (see previous answer). As reported in the Materials and Methods section, the protocol used for the Rotarod tests consisted of two phases in which the rotation speed of the horizontal rotating rod can be constant or accelerating. The tests have been developed to measure motor coordination (accelerating conditions) and motor coordination associated to muscular fatigue (constant speed). As the results of our experiments with the two rotarod set up are very similar, and since it has been already reported

that muscular strength and locomotory endurance are not impaired in CpKO mice (Patel et al., J Neurosci 22: 6578, 2002), thus fatigability is not increased. In order to make more clear this message we removed from the figure 2 the results of the constant speed rotarod, keeping the results of the accelerating speed test, that is more suitable for the detection of motor coordination impairment. The figure legend and text have been accordingly modified. Constant speed rotarod test description has been removed from the materials and methods section, and multivariate analysis were done ex-novo excluding the constant speed rotarod test results. Accordingly also the original figure 8 has been replaced with the new one that inferred conclusions identical to the previous, but also reinforced by the unsupervised detection of two clusters that correspond respectively with the group of CpKO and the group of all the other samples including also Cp-treated KO mice.

Figure 3: A. The exogenous and endogenous images are not convincing, especially in the WT+ Cp. The authors should consider removing this concept from the paper.

Authors' answer: We agree with the Referee that the size and quality of the Western blot image in figure 3A do not easily allow to appreciate the difference in molecular weight of the exogenous and endogenous Cp. However, we believe that the concept is important because it allowed us to distinguish the two kind of molecules and evaluate in the Cp treated WT mice if, and how much, administered Cp entered the brain in a way that is clearly separated/independent from the endogenous Cp expression. Furthermore, this concept with the same slightly difference in molecular weight between human and rodent Cp has already been reported in literature (Hahn P et al. PNAS 101: 13850-55, 2004). A sentence reporting this information and the relative bibliographic reference has been introduced in the revised version of the manuscript ["A similar difference in molecular weight has already been reported between human and rat Cp (Hahn et al., 2004)"].

In order to obtain a better separation between exogenous human Cp and endogenous Cp in treated mice (see also the suggestion from Referee#2 at comment 2) new experiments were performed. In the revised version of the manuscript images obtained from a repetition of the Western blot experiments, were used to substitute the original panel A in figure 3.

In addition, we show here for the Referees only, that when purified Cp is directly added to the brain lysate of the WT mice in doses similar to those quantified in the lysate of the Cp-treated CpKO mice (from 50 to 500 pg), we are able to see the specific signal distinguished from the endogenous Cp signal. Thus, if Cp entered and accumulated in the brain of WT treated mice, at least in amount comparable to CpKO mice, we would have been able to detect it. These observations point toward our hypothesis that administered Cp is not able to cross the barrier systems in WT mice, or if it does, it does it in undetectable amount.

7.5% SDS-PAGE and western blot analysis performed with anti-Cp Ab on: purified human Cp at 50, 150 and 500 pg; brain lysate (30 µg) from original WT mice and WT mice treated for 2 months with Cp (WT+Cp); and brain lysate (30 µg) from original WT mice in which different amount of purified Cp have been directly added in the lysate before electrophoresis. Exogenous administered Cp and endogenous murine Cp are clearly distinguishable (arrows) and exogenous Cp signal is yet visible in the WT lysate when added at 50 pg. *= purified apo-Cp; dash= background signals also visible in the images of the manuscript. Please take into consideration the "smile" effect on the electrophoretic

run.

Figure 3 and 4. Why is there no apparent correlation between the amount of protein and ferroxidase activity across all samples. Please address.

Authors' answer: The apparent lack of correlation between the amount of protein and ferroxidase activity was explained in the Discussion section (bottom page 15 of the original submission) and is dependent on the higher ferroxidase activity reported for the human Cp compared to mouse Cp, which was estimated about 2-6 fold greater depending on the kind of assay used (Gray et al 2009). Therefore, even if in Cp-treated CpKO mice the level of the protein was on average about a half of the Cp expressed by WT mice (ranging from 1/1 to 1/20), there was a full recovery of the ferroxidase activity, which is on average even higher than the activity in WT, and ranged from 2/1 to 1/10.

Less clear is why in WT mice treated with Cp there was an apparent trend of reduction of the ferroxidase activity (see previous answer).

Fig 4) Why is there less CP activity in brain following cp administration to the wt?

Authors' answer: see above answer to the general comment on Results section.

Figure 6d and E. Brown staining is not obvious. Brown stain is very difficult to see on a blue and pink background, Can sections be re stained? There are no scale bars. Remove the sentence images were taken at X40 . The Perls stain does not seem to have worked/

Authors' answer: For image acquisition we used the Nuance® FX multiplex image system (PerkinElmer) that is based on multispectral image acquisition on different wavelength. This is done in order to allow the clear separation of signal from specific color for semi-quantitative analysis, avoiding noisy cross contamination of the signal. The images presented in the original manuscript were the result of a grey-scale recording acquired at different wavelengths that were computationally reassembled in pseudo-color. Thus, the brown/pink staining was not a background but was the translation in false-color of DAB staining of iron accumulation that was not observable in the brain tissue outside of the choroid plexus (see images on the right of the panel D in the original figure 6 and in the new Appendix figure S4). In addition, the signal is not as the expected classical particles staining due to iron deposition. This is because, in this very early phase of the neurological alteration, rather than an iron deposition we expect a diffuse iron accumulation within the cells (see CpKO vs. WT enlarged images in the new figure 6 panel E where intracellular staining is visible in KO but not in WT). This hypothesis is supported by the reported increase of ferritin expression in the choroid plexus of CpKO mice (Rouault et al, 2009) to deal with iron accumulation. Speckled staining has not been reported for iron in the brain of young CpKO mice, before 12-14 months; iron content in younger mice has been measured with non-histochemistry methods (ICP-MS; flame atomic absorption spectrometry). Furthermore, the modified-Perls' staining we used, namely Turnbull method, detects ferrous iron and, in order to increase its sensitivity, the methods include a step of total cell iron reduction (namely by incubation in ammonium sulphide) which in turn allows to detect the total iron contained in cells independently from its original Fe²⁺ or Fe³⁺ status. This should also include for example the iron accumulated within cells in ferritin in response to cell iron dismetabolic overload.

For these reasons the semi-quantitative analysis we have done was based on the ratio of the area of iron/cells staining rather than to evaluate speckle staining.

In order to make clear to the readers, which kind of signals were used for the semi-quantitative evaluation, corresponding to the different staining (modified-Perl's for iron and cresyl violet for cells), the image enlargement insertions shown in the original fig. 6D have been further magnified as independent panels in the new figure 6; in addition, the corresponding gray scale images originally acquired for the different wavelength have been introduced. Accordingly, the Figure legend has been modified as follow: "Iron staining (in brown) in choroid plexus epithelium, cells are counterstained in blue. Images acquired with Nuance® FX multiplex image system (PerkinElmer) at different wavelength were computationally reassembled in pseudo-color. Enlargement of the area indicated in left panel insertions and their corresponding gray scale images

originally acquired for different wavelength are also shown. Scale bars= 50 μm and 10 μm for low and high magnification, respectively.”.

Scale bars have been added in the images and the sentence "images were taken at x40" has been substituted with "Scale bars= 50 μm and 10 μm for low and high magnification, respectively”.

Figure 7. The loss of cells in the 10 month KO is spectacular, do you have images at 8 and 9 months.

Authors' answer: We agree that to have this kind of information it would be very interesting, but we don't have any mice sacrificed before and during the Cp-treatment.

Figure 8 Reviewer suggests removing this figure it distracts and doesn't add to the argument.

Authors' answer: See above answer on computational analysis issue.

Appendix

Change title "Evaluation of total brain proteins oxidation."

Please also address why there were no protein carbonyls in the KO group.

Authors' answer: Title has been changed as follow: "Evaluation of protein carbonylation in brain "

Protein carbonylation of the KO mice, with the exception of one animal, was comparable to that of the other groups. We already addressed this result in the discussion section of the original manuscript reporting the observations that both brain protein and lipid oxidation was seen in older mice (Patel et al, 2002 J Neurosci; Hineno et al. 2011 Neurochem Res); therefore, in our KO animals the oxidative stress/damage was not yet occurring. The original sentence quoted is: "Thus, the oxidative damage consequent to iron accumulation, reported as lipid and protein oxidation in older mice (Patel et al, 2002), is not yet occurring, as inferred by the similar level of protein carbonylation observed in CpKO and WT mice. Hineno et al. also showed a lack of protein and lipid peroxidation in CpKO mice of 14 months (Hineno et al, 2011). The indication that oxidative stress is not yet occurring in these mice could be an advantage from the therapeutic point of view. Indeed, the pro-oxidant environment is able to induce Cp modifications....."

Delete the Appendix S3: The Perls stain does not seem to have worked. There is no scale bar on the figures. Remove sentence "Images were taken at x20". The WT has dirt in the image.

Authors' answer: see previous answer to Figure 6 comment.

In order to make clear the different signals corresponding to the different staining (Perls for iron and cresyl violet for cells) the computationally assembled pseudo-color images and their original gray scale images, acquired for the different wavelength, have been introduced in the new Appendix S4 figure replacing the original light microscopy images (ex-S3 appendix figure). In addition, enlargement of the region including microvascular endothelia of the images acquired at wavelength for cells and iron staining have been included in the figure. Accordingly, the explanatory text and figure legend has been modified as follow: "Images were acquired with Nuance® FX multiplex image system (PerkinElmer) at different wavelength corresponding to the chromophore emission."..."Bottom panels show enlargement of the regions including endothelia microvasculature. Images acquired at different wavelength were computationally reassembled in pseudo-color (iron= brown; cells= blue), and their corresponding gray scale images are also shown."

Scale bars have been added and the sentence "images were taken at x20" has been substituted with "Scale bars= 50 μm and 25 μm for low and high magnification, respectively”. The usage of the computationally assembled pseudo-color images clean up the artefact on the WT image being its signal not on the wavelengths acquired for signal staining.

The Reviewer suggested to delete appendix S4. "Brain" TH is not a valid assessment of neuronal number. Cpu or Sn th is used to assess loss of dopamine in motor control circuits but whole brain TH is a very poor measure. The original paper by Prof David phenotyping the mouse photographed the nigra and showed loss of cells.

Authors' answer: Appendix S4 has been removed (see above).

Delete Multivariate analysis. S1

Authors' answer: See above answer on computational analysis issue.

Referee #2 (Remarks for Author):

The paper by Zanardi et al reports the approach of enzyme replacement therapy in a knock-out mouse model for ceruloplasmin (CP), a protein playing key role in iron homeostasis. In human, mutations of CP cause aceruloplasminemia a disorder characterized by iron accumulation in liver, pancreas and brain causing diabetes, retinal degeneration and progressive neurodegeneration. The paper is potentially relevant despite several weaknesses are present and the data are not always clearly reported.

Specifically, a first observation is that up to the results session it is not evident that the protein used for ERT was of human origin. I think this crucial information should be reported in the abstract.

Authors' answer: As requested the human origin of the Cp protein used for ERT has now been indicated in the abstract and in the last paragraph of the introduction section.

As concerning the results I would like to comment on the following experiments:

1) In Figure 1 panel A there are only two animals /group while the treated animals were 3. In addition, only one untreated animal is present and no WT animals are reported just to make the comparison of the results easy to interpret.

Authors' answer: To avoid the risk of technical pitfalls due to the fact that a homogenous group of samples (e.g. all animals at days 7 and 10) could be loaded on a different gel, the samples from animals at different time were split onto two different gels and run in parallel. Thus the third mouse from each time group of treated mice together with the other two untreated KO animals were loaded on a different gel. Only one representative gel showing the heterogeneity of the Cp levels was presented in the figure. Here is reported for revision purposes the image of the other complementary gel showing the remaining samples for the specific experiments presented in the figure of the manuscript.

Regarding the WT animals, these were not investigated in the pharmacokinetics analysis because those were experiments aimed to set up the experimental conditions necessary to confirm the capability of administered Cp to enter the brain and to define a frequency of Cp-administration useful to allow a protein accumulation in the brain of CpKO mice. Moreover, at the beginning of the project, when pharmacokinetics was done, we were not yet aware that endogenous mouse Cp and the exogenous human Cp can be distinguished by different molecular weight. Therefore, in order to have clear interpretable images the experiments were done only with CpKO mice.

Moreover, at day 5 the WB shows faint bands while the evaluation of Cp expression in panel B, is higher than that at day 1 or 2 where the WB shows an intense band. How the Authors explain these contradictory results?

Authors' answer: Due to the reported heterogeneity of the Cp signal found in the different mice (see Results section of the manuscript) and the technical choice of running samples split in different gels (see above) the Western blot image is only qualitative/representative of the Cp levels. In fact, in the second gel, showing the Cp level of the third animal for each time-group (see image above), at day 5 there is a stronger signal that explains the apparent contradiction, and justify why the average signal is high. Moreover, the quantification is the result of the average evaluation of Cp signal in each mouse obtained from technical replicates of the gels.

2) In Figure 3 the WB in panel A is not very clear. I would suggest either to perform a less exposure to avoid saturation of signals or to run the gel for more time in order to obtain a better separation of the proteins and demonstrated the different MW of the human and mouse Cp protein. Is the Cp on the right the purified human protein? This should be specified in the legend of the figure. Moreover, the Authors claim that the externally administered CP was not able to cross the BBB in WT mice. Again this is very difficult to say from the presented picture. Also in panel D, especially the right part, it is difficult to identify the band.

Authors' answer: See answer to Figure 3 comment of Referee#1 where the issue of endogenous and exogenous Cp is addressed together with the concern on the inability of administered Cp to cross brain barrier systems in WT mice. A statement on the purified human Cp loaded as control in the gel has been added in the figure legend of the revised manuscript. " Purified Cp was run alone as control for semi-quantitative evaluation (Cp, lanes on the right)"

Regarding the quality of panel D images, we faced with technical difficulties, for example low sensitivity of lectins in comparison to antibodies, or difficulties encountered with neuraminidase protocol set up. In fact, 18 hrs incubation at 37 °C is required for sialic acid removal in organ lysate, thus, some protein degradation might have occurred even in the presence of protease inhibitor. Moreover, the restricted gel loading capacity did not allow us to have enough brain lysate material as to contain exogenous Cp level suitable for a good Western blot signal.

We performed new experiments and the resulting images were more clear and convincing on the sialylated status of the administered Cp. This supports the hypothesis that endothelial transcytosis is not the prevalent mechanism used by Cp to enter the brain of CpKO mice. In the figure 3 of the revised manuscript panel D images have been substituted, the panel enlarged and contrast modified as to better identify the bands. The following sentence has been added in the figure legends. "Double arrows indicated sialylated Cp; dash indicates anti-Cp reactivity after sialic acid removal. In panel D, image contrast has been homogeneously modified to better identify the bands."

3) In Figure 4A it is reported the ferroxidase activity in the brain of mice. While it is clear the effect of Cp expression in KO animals, it is not clear why WT mice injected with Cp showed a reduced ferroxidase activity. This should be explained and motivated.

Authors' answer: see answer to Referee#1

4) Despite the fact that neurodegeneration is the major cause of disability in human (and mice), the Authors should also consider iron accumulation in liver and pancreas as part of the disease spectrum. I would suggest to measure iron in liver of the Cp KO mice and to evaluate the effect of Cp administration. This is particularly important in evaluating the potential of ERT therapy in humans. In fact, Authors describe the possibility of a "preventive therapy" in a period in which neurological symptoms are not evident.

Which are the first signs of the disease in humans that could be monitored to start a preventive therapy?

Authors' answer: It would be of great interest to investigate the effects of ERT on systemic alteration of Acp, but this was out of the scope of this project. Our study was focused on the neurodegenerative aspect of this rare genetic disease, and systemic alterations will be the aim for

future extended studies. Nevertheless, as suggested we measured the liver iron in CpKO mice and the effect Cp treatment. We found a significant higher iron concentration in the CpKO mice compared to WT that was reduced after Cp administration with a trend that, however, did not reach statistical significance. This might be due to the fact that iron content in liver of CpKO mice was much greater than in brain (about 1600 vs. 11 $\mu\text{g/g}$) suggesting that the amount of the administered Cp could be enough to efficiently reduce iron accumulation in the brain but not in the liver. In addition, the faster clearance of exogenous Cp from plasma compared to the brain may partly explain such a difference, too. Longer Cp treatment or increased dose of administered Cp might lead to full recovery of physiological conditions also in liver. The results of these new experiments have been added in the revised version of the manuscript in the Appendix section (new Appendix Figure S3). These data were quoted in the Results section and commented in the Discussion as follow: Results section, "Similar to the brain, we found a significantly higher iron concentration in the liver of CpKO mice compared to WT (1535 ± 341.3 vs. 517.1 ± 164.4 $\mu\text{g/g}$ dry tissue), which appeared to be reduced after Cp administration (1208 ± 241.8 $\mu\text{g/g}$ dry tissue), however, did not reach statistical significance (Appendix Fig. S3)."; Discussion section, "We did not investigate in detail the effect of the Cp ERT on the systemic deficiencies in Cp-KO mice, but we found that the significant high iron concentration detected in the liver of CpKO mice showed a trend in reduction after Cp treatment. The lack of a full effect might be due to the amount of iron detected in the liver of CpKO mice that was much greater than iron detected in brain; thus, the amount of the administered Cp reaching the liver is probably not enough to efficiently reduce iron accumulation. Longer Cp treatment or increased dose of administered Cp might lead to full recovery of physiological conditions also in liver".

Moreover, in order to underline the potential efficacy of Cp ERT also on systemic alteration of Acp patients, we added in the Discussion of the new version a sentence reporting the effect that administration FFP containing Cp showed on iron homeostasis in Acp patients ("Moreover, a transient rescue of serum iron mobilized from stores, has also been shown in Acp patients after administration of FFP containing ceruloplasmin (Logan et al, 1994; Yonekawa et al, 1999).").

As for the last point raised by the referee, the two major recent studies recapitulating the clinical manifestation in a large number of aceruloplasminemia patients reported, although with different percentages, that the first signs of Acp are diabetes mellitus and microcytic anemia that is refractory to iron administration therapy, often both occurring concomitantly (Vroegindeweij et al., 2015; Kono & Miyajima, 2015). In a series of seven patients followed at the Centre for Disorders of Iron Metabolism, ASST-S.Gerardo Hospital, Monza, Italy by one of the co-author (AP), all had microcytosis, 86% had mild-moderate anemia, low serum iron and transferrin saturation, and high serum ferritin, while 57% had diabetes or glucose intolerance, at diagnosis. Thus, given the phenotype heterogeneity and the non-specificity of early symptoms, Acp diagnosis must be confirmed by measurement of the level and functional activity of plasma ceruloplasmin and molecular genetic analysis.

5) The Authors describe loss of Purkinje cells as responsible for the movement disorder in mice. Is the same phenotype present in humans? Is cerebellar atrophy a common feature in aceruloplasminemia? This information should be added.

Authors' answer: A marked loss of Purkinje cells in cerebellar cortex is included in the post-mortem laboratory pathological diagnosis criteria reported in the GeneReviews® (NCBI Bookshelf, National Library of Medicine, NHI; Miyajima H. Aceruloplasminemia. In GeneReviews®, Pagon RA, Adam MP, Ardinger HH, Wallace SE, Amemiya A, Bean LH, Bird TD, Fong CT, Mefford HC, Smith RJH et al. Eds. Seattle, WA. 1993 update November 2015). Even though for obvious reasons not all described Acp patients have been investigated for loss of Purkinje cells, the cerebellar ataxia is reported in 40-50% of Acp cases.

These information have been introduced in the discussion section of the revised manuscript as follows: "Interestingly, a marked loss of Purkinje cells has been reported in human Acp, where the cerebellum appeared slightly shrunken and patients developed cerebellar ataxia (Kono, 2012; Miyajima, 2003; Miyajima et al. 2001; Morita et al. 1995). Two meta-analysis of 55 and 71 cases described worldwide up to the year 2015, showed cerebellar ataxia in the 41.5% and 48.2% of patients, respectively (Vroegindeweij et al., 2015; Miyajima 1993). These reports indicated that in about 50% of Acp patients the cerebellar phenotype may be similar to that of the preclinical model suggesting that Cp replacement therapy might be efficacious also in human."

6) There is evidence for iron accumulation in choroid plexus epithelium in CpKO mice that is also responsible for the success of the ERT with the human Cp protein, which can permeate thus reaching the brain.

However, in aceruloplasminemia iron accumulation is in other brain region and there are no evidence that the same phenomenon could guarantee the success of a therapy in humans. Could the Authors comment on this?

Authors' answer: we agree with the Reviewer that till now no evidences of iron accumulation in the choroid plexus have been reported in Acp patients and that this could be an important issue for the success of the ERT in the light of our hypothesis of barrier systems leakiness. However, we think that this may be part of the intrinsic limitations of the use of preclinical models. We believe that the important evidence we obtained from our study is that when Cp is replaced within the brain it displays a therapeutic effect. This could be true also in human Acp. Therefore, even if the barrier systems leakiness will not be exploitable in human Acp, it would be worth enough to explore alternative systems for Cp delivery within the CNS (e.g. engineered Cp suitable for receptor mediated transcytosis, or nanoparticles, etc.) or take into consideration a local production of the protein inducing an intra-ventricular Cp expression by ependymal cells directed gene-therapy.

Sentences underlining the difference between preclinical model and human Acp, and addressing possible alternative strategies for Cp delivery in CNS have been added in the Discussion of the revised manuscript ["However, there are no reports on iron accumulation in the choroid plexus in Acp patients, likely due to little attention paid to this brain region. On the other hand, iron deposition was shown in perivascular astrocytes surrounding endothelial cells of brain microvasculature, suggesting a possible dysregulation of iron homeostasis in these structures (Kaneko et al, 2002; Oide et al, 2006; Gonzalez-Cuyar et al, 2008)." and "This, can be examined even if the leakiness of B-CSF-B system would not be exploitable in human Acp, due to the lack of evidence supporting iron accumulation in choroid plexus of patients. Indeed, it would be worth to explore alternative systems for Cp delivery to the CNS or an intra-ventricular Cp expression (e.g. engineered Cp suitable for receptor mediated transcytosis, nanoparticles, ependymal cells directed gene-therapy, etc.)"].

Referee #3 (Comments on Novelty/Model System for Author):

The Reviewer stated: "The manuscript was well written, the citation of the references is appropriate."; also "These findings are very important for treatment of aceruloplasminemia patients."

The Reviewer also stated: There are some issues I believe the authors need to address.

Combination therapy with fresh frozen plasma to replenish the blood ceruloplasmin levels, and thereafter the administration of desferrioxamine to deplete iron stores, showed an improvement in the neurological symptoms. However, after receiving long-term treatment with fresh frozen plasma by intravenous infusion once a week for two years, a brain MRI evaluation demonstrated improvement in the low intensity areas in the basal ganglia, but showed little effect on neurological symptoms. The ceruloplasmin replacement therapeutic approach remains controversial in patients with aceruloplasminemia. Iron deposition is more severe in the endothelial cells of the microvasculature than in choroid plexus in aceruloplasminemia brain. Distribution of iron deposition is different between aceruloplasminemia patients and ceruloplasmin-knock-out mice. It isn't necessarily true that ceruloplasmin replacement therapy ameliorate neurological symptoms in all patients with aceruloplasminemia.

Authors' answer: (see answer to point 1 below)

Referee #3 (Remarks for Author):

Aceruloplasminemia is a slowly progressive neurodegenerative disorder, and an early diagnosis and early treatment of patients are important issues. To reduce iron accumulation, iron chelation therapy is effective for reducing the hepatic and pancreatic iron overload and improving diabetes,

but not effective for neurological symptoms. In this paper, intraperitoneal-administered human ceruloplasmin was able to enter the brain inducing replacement of the protein levels and rescue of ferroxidase activity of ceruloplasmin-knock-out mice. Ceruloplasmin-treated mice showed amelioration of motor incoordination associated with diminished loss of Purkinje neurons and reduced brain iron deposition. These findings are very important for treatment of aceruloplasminemia patients

There are some issues I believe the authors need to address.

1. Combination therapy with fresh frozen plasma to replenish the blood ceruloplasmin levels, and thereafter the administration of desferrioxamine to deplete iron stores, showed an improvement in the neurological symptoms. However, after receiving long-term treatment with fresh frozen plasma by intravenous infusion once a week for two years, a brain MRI evaluation demonstrated improvement in the low intensity areas in the basal ganglia, but showed little effect on neurological symptoms. The ceruloplasmin replacement therapeutic approach remains controversial in patients with aceruloplasminemia. Iron deposition is more severe in the endothelial cells of the microvasculature than in choroid plexus in aceruloplasminemia brain. Distribution of iron deposition is different between aceruloplasminemia patients and ceruloplasmin-knock-out mice. It isn't necessarily true that ceruloplasmin replacement therapy ameliorate neurological symptoms in all patients with aceruloplasminemia.

Authors' answer: To our knowledge, up to now only four published reports showed results on the FFP administration alone or in combination with deferoxamine (Logan et al. Q J Med 87: 663, 1994; Yonekawa et al. Eur Neurol 42: 57, 1999; Skidmore et al. J Neurol Neurosurg Psychiatry 79: 467, 2008; Poli et al. Neurol Sci 38: 357, 2017) but, to our best knowledge, none of them reported data obtained after two years of FFP therapy. Are they unpublished results? As we do not have access to these data it is difficult to properly address the Reviewer's comment. Available data are scanty and do not allow one to draw any conclusion. The present paper is the first evidence that exogenous Cp has positive effects at biochemical, histological, neurological levels in Acp.

Our comment on the quoted information is that the efficacy may depend on the neurological damage that patients already have before the FFP administration. We would expect a reduction of iron deposition but only a stop or slowdown of neuronal loss. In other words, a regeneration of already lost neuronal tissue is not an option, while removal of iron deposition is feasible. We expected that the same occurred in our CpKO mice that were treated when neurological symptoms were just beginning; for example the Cp-treatment was able to reduce the progressive loss of Purkinje cells, but is not able to regenerate the cells already lost. This is an important distinction. We suggest that Cp-treatment can prevent neuronal loss not regenerated lost neurons. This could be of value clinically as patients can be put on ERT when they are first diagnosed.

We recognize the limits that preclinical models have, as not all the mouse disease features can be identical to those of human. However, we believe that the important evidence we obtained from our study showing that Cp replaced within the brain displays a therapeutic effect, is worth exploring in human Acp patients, even with the risk that it will may not be as effective (see also answer to point 6 of Reviewer#2).

2. The authors described that the administered ceruloplasmin that reached the brain entered the central nervous system by a mechanism different than endothelial transcytosis. The authors should explain blood-brain barriers permeability and leakage mechanism in detail.

Authors' answer: From the biochemical evidence that the administered Cp found in the brain of CpKO mice was still sialylated (see also the improvement in the quality of the results showed in the revised version of the manuscript; answer to point 2, Reviewer #2), we hypothesized that endothelial transcytosis was not the prevalent mechanism through which Cp enter the brain of CpKO mice, at least at early neurological symptoms occurring in 10 months old mice. We can't completely rule out transcytosis contribution, but, if there is, it should be minimal. The sentence referring to this issue in the original Discussion has been modified as follow: "Nevertheless, our results did not support endothelial transcytosis as a prevalent mechanism of exogenous Cp entry into the brain, at least in the early stage of the neurological phase of the disease."

Therefore, as alternative mechanisms we hypothesized that Cp entry to CNS was likely due to barrier systems leakage. Iron accumulation found by histochemistry analysis lead us to hypothesize

that leakiness of the B-CSF-B might have major chance to occur than leakiness of the BBB, at least in the early stage of the neurological phase of the disease. However, from the experimental point of view it was not possible to get more information, as we underlined in the discussion of the original version of the manuscript. This will be an option to assess for future studies. Therefore, at the moment is difficult to add more details on the mechanism of permeability and leakage of the brain barrier systems in the preclinical model. (see also answer to point 6 of Reviewer#2)

The sentence, already present in the discussion, quoting the possible future studies addressed to understand the barriers permeability, has been expanded in the revised version introducing also the concept of cellular model studies ("Further studies on *in vivo* Cp biodistribution (e.g. by PET analysis of ⁶⁴Cu-labelled Cp administration) and *in vitro* in CpKO-cellular model of brain barrier systems, will be useful for the investigation of the hypothesised barriers permeability and leakage mechanism.")

3. The authors described that no alterations in copper and zinc expression have been reported in aceruloplasminemia patients. In aceruloplasminemia brains, however, massive iron deposition was observed in the brain associated with low zinc content and high copper content.

Authors' answer: we apologize for the lack of accuracy of our statement that was referring to Cu and Zn content in the brain only. In the revised manuscript the sentence has been modified and expanded as follow: "The copper content in the brain of Acp patients has been evaluated in few cases and was found not altered (Morita et al, 1995; Yoshida et al, 1995). However a recent report showed a copper accumulation in the iron-rich particles in the brain of three Acp patients (Yoshida et al, 2017), and a copper level elevation in two patients was mentioned in (Miyajima, 2015). On the contrary, a tendency to decrease of zinc concentrations in the brain of two Acp patients has been reported (Miyajima, 2015)."

2nd Editorial Decision

26 October 2017

Thank you for the submission of your revised manuscript to EMBO Molecular Medicine. We have now received the enclosed reports from the referees that were asked to re-assess it. As you will see the reviewers are now supportive and I am pleased to inform you that we will be able to accept your manuscript pending editorial amendments.

***** Reviewer's comments *****

Referee #1 (Remarks for Author):

While this reviewer disagrees with some the arguments the authors present a well argued scientific discussion. The Authors have addressed this reviewer's comments. This is a very important article that may lead to the clinic.

Referee #2 (Remarks for Author):

The Authors have performed additional experiments, modified figures and text according to the requested modification. They also improved significantly the discussion.

Referee #3 (Remarks for Author):

The revised manuscript was well written, the citation of the references is appropriate. This paper is important for treatment of aceruloplasminemia patients.

Corresponding Author Name: Massimo ALESSIO

Journal Submitted to: EMBO - Molecular Medicine

Manuscript Number: EMM-2017-08361